# Genomic loci mispositioning in *Tmem120a* knockout mice yields latent lipodystrophy

Rafal Czapiewski [1], Dzmitry G. Batrakou[1], Jose I. de las Heras[1], Roderick N. Carter[2], Aishwarya Sivakumar[1], Magdalena Sliwinska[1], Charles R. Dixon [1,3], Shaun Webb[3], Giovanna Lattanzi[4,5], Nicholas M. Morton [2] & Eric C. Schirmer [1✉]

Little is known about how the observed fat-specific pattern of 3D-spatial genome organisation is established. Here we report that adipocyte-specific knockout of the gene encoding nuclear envelope transmembrane protein Tmem120a disrupts fat genome organisation, thus causing a lipodystrophy syndrome. Tmem120a deficiency broadly suppresses lipid metabolism pathway gene expression and induces myogenic gene expression by repositioning genes, enhancers and miRNA-encoding loci between the nuclear periphery and interior. Tmem120a$^{-/-}$ mice, particularly females, exhibit a lipodystrophy syndrome similar to human familial partial lipodystrophy FPLD2, with profound insulin resistance and metabolic defects that manifest upon exposure to an obesogenic diet. Interestingly, similar genome organisation defects occurred in cells from FPLD2 patients that harbour nuclear envelope protein encoding *LMNA* mutations. Our data indicate TMEM120A genome organisation functions affect many adipose functions and its loss may yield adiposity spectrum disorders, including a miRNA-based mechanism that could explain muscle hypertrophy in human lipodystrophy.

[1] Institute of Cell Biology, University of Edinburgh, Edinburgh EH9 3BF, UK. [2] Molecular Metabolism Group, University/BHF Centre for Cardiovascular Science, Queen's Medical Research Institute, University of Edinburgh, Edinburgh EH16 4TJ, UK. [3] Wellcome Centre for Cell Biology, University of Edinburgh, Edinburgh EH9 3BF, UK. [4] CNR - National Research Council of Italy, Institute of Molecular Genetics "Luigi Luca Cavalli-Sforza", Unit of Bologna, Bologna 40136, Italy. [5] IRCCS, Istituto Ortopedico Rizzoli, Bologna 40136, Italy. ✉email: e.schirmer@ed.ac.uk

Human diseases linked to defects in adipose tissue and metabolism cover a wide range, the extremes of which being severe obesity and lipodystrophy. Nonetheless, both these extremes of adiposity lead to insulin resistance and metabolic disease. These disorders often arise from rare protein-coding mutations causing adipose tissue dysfunction; however, genome association studies also point to more complex mechanisms[1–3]. For example mutations in remote enhancers that disrupt adipocyte-specific gene expression are now linked to adiposity pathologies[1–3] and alterations in genome organisation that block enhancer-gene interactions are demonstrated to disrupt tissue development leading to disease[4,5]. Despite these clear links to genome organisation, the proteins controlling adipocyte-specific genome organisation have yet to be identified. Doing so could yield significant benefits to understanding and treating adipose disease because genome organisation contributes to tissue differentiation and gene expression through directing genes to active/inactive regions within the nucleus and enabling enhancer-gene contacts.

The nuclear envelope (NE) actively directs radial genome organisation, releasing for activation genes needed for tissue differentiation and function while recruiting for repression genes antagonistic to tissue function[6]. Importantly, radial genome organisation is disrupted in diseases linked to NE proteins such as A-type lamins. Mutations in the *LMNA* gene that encodes A-type lamins (principally lamins A and C) cause familial partial lipodystrophy of the Dunnigan-type (FPLD2)[7,8], a more severe lipodystrophy[9], several metabolic syndromes[10,11], and type 2 diabetes[12,13]. The relevance of genome organisation for adiposity was previously indicated by findings that critical pro-adipogenic genes *FABP4*, *PPARG*, *CREB* and *CEBPB* reposition from the NE to nuclear interior in adipogenesis[14]. Lamin A was the first NE protein linked to adipogenesis, with >1,000 genes under NE/lamin A positional regulation in adipocytes[15,16]. However, lamin A is widely expressed while several tissue-specific NE transmembrane proteins (NETs) direct specific gene repositioning during development of the tissues where they are expressed[17–19]. Of these, TMEM120A (called NET29), which notably we found in both the nuclear envelope and ER[20], is highly expressed in adipose tissue[21].

In humans, the mRNA level of *TMEM120A* was ~25% lower in adipose tissue of obese Arizona Pima Native Americans[22] and 40% lower in obesity with diabetes in Asian Indians (data obtained from GEO dataset, GDS3665). Both studies showed stronger effects in women. In the mouse 3T3-L1 adipogenesis tissue culture system, *Tmem120a* gene knockdown in pre-adipocytes reduced adipogenesis and lipid accumulation ~30% and combined knockdown with its paralog *Tmem120b* reduced it ~60%[21]. Notably, TMEM120A re-expression in double knock-down 3T3-L1 cells elicited superior adipogenic rescue than TMEM120B, suggesting a dominant role of TMEM120A in adipogenesis. Intriguingly, *Tmem120a* is also upregulated by cold-exposure in WAT of mice[23] and by treatment of 3T3-L1 mouse cells with rosiglitazone, an agonist of the master adipogenic transcription factor PPARγ, that improves insulin sensitivity in diabetes[24]. Finally, brain is one of the few other tissues expressing low *Tmem120a* mRNA and antibodies detected a larger mass Tmem120a protein in mouse brain, suggesting a different isoform between adipose tissue and the nervous system[21]. *Tmem120A* was one of the genes with most altered expression in hypothalamus of chickens genetically selected for fat accumulation[25]. A different study suggested that TMEM120A (called there TACAN) is involved in nociception in brain[26]; however, this presumed role as a mechanosensing channel was challenged by three independent studies published this year[27–29].

Here we investigated the impact of adipocyte-specific *Tmem120a* gene disruption on genome organisation and fat metabolism in mice in vivo. We find that loss of Tmem120a leads to a distinct latent lipodystrophy pathology, similar to FPLD2. At the cellular level, we find that loss of Tmem120a alters positioning of multiple genes, enhancers, and miRNA-encoding loci resulting in multiple gene expression defects, particularly reducing expression of fat metabolism genes and activating/derepressing muscle genes. This suggests a disease pathomechanism in which the altered genome organisation causes changes in many genes to affect adiposity. Importantly, this is the demonstration that miRNA-encoding loci are under NE positional regulation and this, together with several targets of the altered miRNAs being upregulated muscle genes, suggests that lipodystrophy-associated skeletal muscle hypertrophy could be paracrine mediated.

## Results

**Adipocyte–specific *Tmem120a*[−/−] mice exhibit reduced white adipose but increased brown adipose fat accumulation on a high fat diet.** Given that *Tmem120a* knockdown in 3T3-L1 in vitro adipogenesis reduced expression of several genes central to adipocyte metabolism[21], we sought to determine effects of Tmem120a loss on adiposity and metabolism in mice. Mating a conditional knockout line with exons 2 and 3 floxed (*Tmem120a*[fl/fl]) to Adiponectin-Cre (Adip[Cre/+]) mice[30–32] generated an adipocyte–specific knockout (*Ad-Tmem120a*[−/−]; Supplementary Fig. 1a–k). Notably, the Tmem120a knockout had no effect on levels of its paralog Tmem120b (Supplementary Fig. 1j). Littermates, homozygotes *Tmem120a*[fl/fl] without the Adip-Cre allele were used as a control for all experiments. Adip[Cre/+] deletor mice were used because it is more specific for adipose tissue than the alternate Ap2[Cre] mice that express Cre in brain[32,33], which was important since a larger Tmem120a variant is expressed in brain. Accordingly, we confirmed that Tmem120a was still expressed in brain of the *Ad-Tmem120a*[−/−] mice (Supplementary Fig. 1k). We anticipate that early differentiation functions of Tmem120a will still occur since it is expressed 2–4 days earlier than adiponectin[21]; so our use of Adip[Cre/+] should preferentially highlight Tmem120a metabolic functions in mature adipocytes.

Mice on standard diet (13.5% fat, 43% carbohydrates, 29% protein) showed sex specific differences in body weight. By 20 weeks of age female *Ad-Tmem120a*[−/−] animals began to weigh significantly less than controls (Supplementary Fig. 2a). Interestingly, a difference in male body weight, although modest, was observed only between weaning and young adulthood (weeks 4-8; Supplementary Fig. 2b). This sex difference may in part be attributed to ~4-fold lower levels of Tmem120a expressed in male mice compared to female mice (Supplementary Fig. 1h, j). To test if the body weight differences were exaggerated by different caloric loads, mice raised initially on a standard diet were switched to either a low-fat diet (LFD; 10.5% fat, 73.1% corn starch, 16.4% protein) or a high-fat diet (HFD; 58% fat, 25.5% sucrose, 16.4% protein) at 12 weeks of age. Body weight was measured weekly, revealing greatly reduced weight gain specifically on HFD–fed female *Ad-Tmem120a*[−/−] mice compared to littermate controls (Fig. 1a, b). At 9–10 weeks post-HFD, a peak difference of ~25% total body weight was observed that corresponded to >50% less weight gained in the knockout. There was no difference in food intake on HFD between genotypes (Fig. 1c) suggesting altered energy expenditure most likely accounts for the difference in the gain of body weight. Body weight effects of altered caloric loads were also sex-specific with no difference between male *Ad-Tmem120a*[−/−] mice and control littermates on either LFD or HFD (Supplementary Fig. 2c).

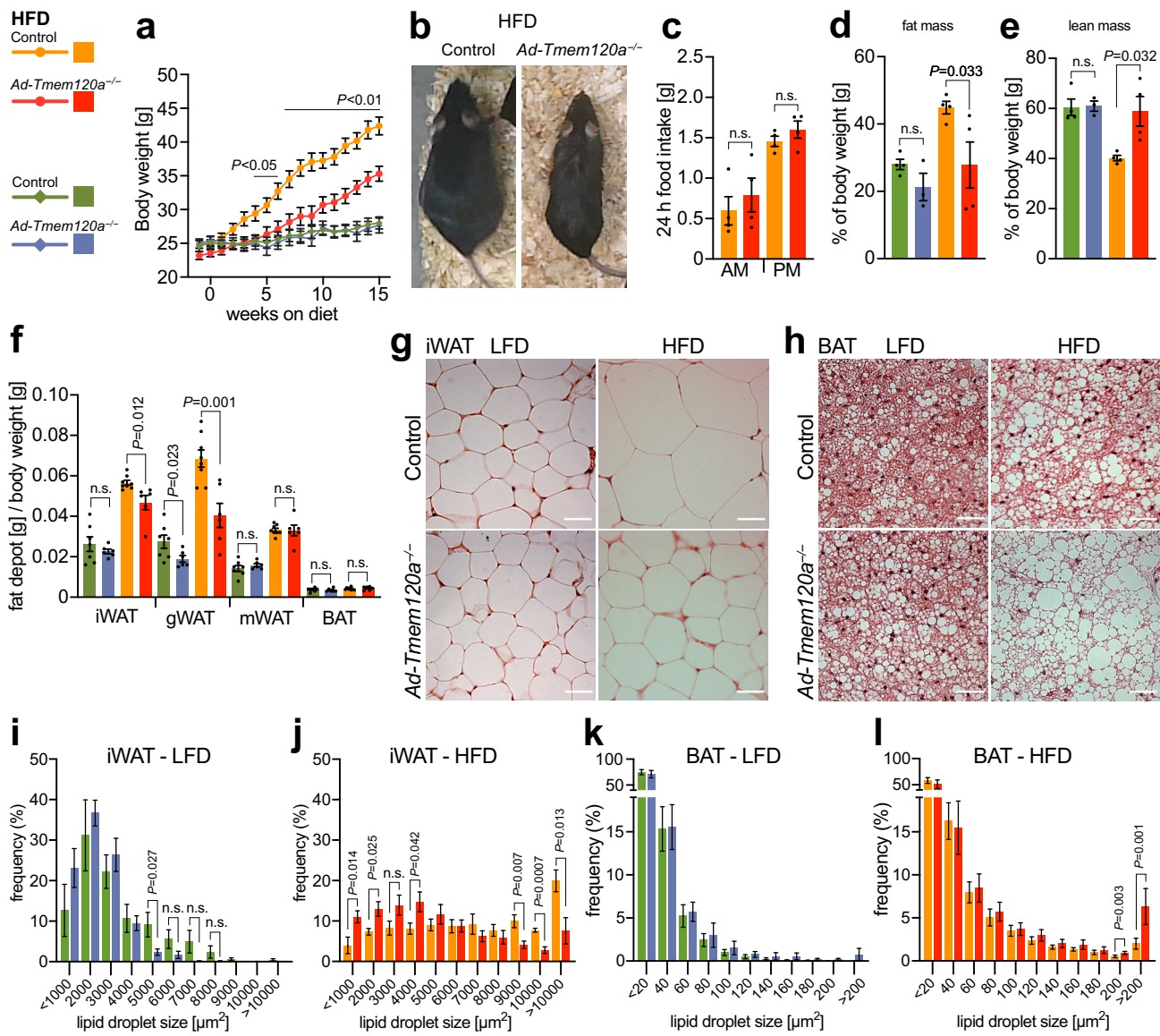

**Fig. 1 Ad-Tmem120a−/− mice have reduced fat accumulation in WAT while fat is increased in BAT. a** Female body weight on high-fat (HFD) vs low-fat (LFD) diet. Data from two separate experiments, shown as mean ± SEM, control LFD n = 9, *Ad-Tmem120a−/−* LFD n = 8, control HFD n = 10, *Ad-Tmem120a−/−* n = 10, P values for each time point are available in Source Data file. **b** Representative images of 6-month-old female mice, control and *Ad-Tmem120a−/−* mice after 10 weeks on HFD. **c** 24 h food intake by females on HFD measured during calorimetry experiment; shown as mean ± SEM (n = 4 for each genotype). Female mice body composition measured by TD-NMR after 10 weeks of LFD and HFD, fat mass (**d**) and lean mass (**e**) are shown as percent of whole-body weight; shown as mean ± SEM (each group n = 4). **f** Whole body mass-normalised weight of different adipose tissue depots: inguinal-subcutaneous (iWAT), gonadal (gWAT), mesenteric (mWAT) and interscapular BAT, shown as mean ± SEM, control LFD n = 7, *Ad-Tmem120a−/−* LFD n = 6, control HFD n = 7, *Ad-Tmem120a−/−* n = 6. Images of haematoxylin and eosin stained iWAT (**g**) and interscapular BAT (**h**), from control and *Ad-Tmem120a−/−* female mice on HFD vs LFD, scale bars 40 μm. **i–l** Quantification of lipid droplet size in iWAT and BAT form control and *Ad-Tmem120a−/−* females on LFD and HFD, the sizes of the lipid droplets were assigned to size bins and are shown as frequency, area in μm², shown as mean ± SEM control LFD n = 7, *Ad-Tmem120a−/−* LFD n = 6, control HFD n = 7, *Ad-Tmem120a−/−* n = 6, P values were calculated by unpaired, two-sided Student's *t* test. See also Supplementary Fig. 1. Source data are provided as a Source Data file.

TD-NMR body composition analysis showed that weight differences were driven by a >40% reduction in gained fat mass using normalised data for *Ad-Tmem120a−/−* females compared to control littermates (Fig. 1d) that was even stronger using non-normalised data (Supplementary Fig. 2d). This difference was partially offset by an increase in lean mass, limiting the difference in total body weight gained to ~25% (Fig. 1e, normalised; Supplementary Fig. 2e, non-normalised). *Ad-Tmem120a* loss had no effect on males either for whole fat and lean mass (Supplementary Fig. 2f, g) or for individual fat depots (Supplementary Fig. 2h). Reduced fat accumulation in females was also

depot dependent. After normalising isolated depot weight to total body weight, gonadal WAT exhibited the largest difference in gained fat mass (>40%) followed by subcutaneous inguinal WAT (iWAT; ~20%) (Fig. 1f).

To test if relative lack of fat mass accumulation in *Ad-Tmem120a−/−* females on HFD was accounted by reduced cell numbers (hypoplasia) or reduced fat accumulation (atrophy), we visually investigated fat stores and quantified cell sizes. iWAT had similar morphology and appearance for droplet/adipocyte numbers per field between *Ad-Tmem120a−/−* and control littermates on LFD indicating no loss in generation of fat cells

during differentiation (Fig. 1g). However, on HFD the size of fat droplets in iWAT cells visibly increased in control littermates compared to Ad-Tmem120a$^{-/-}$ mice (Fig. 1g). Interestingly, this situation was inverted for BAT where fat droplet size per cell greatly increased in Ad-Tmem120a$^{-/-}$ mice compared to control littermates on HFD (Fig. 1h). Quantification revealed ~2.5-fold increase for iWAT lipid droplets >10,000 μm$^2$ in size specifically for control littermates on HFD (Fig. 1i, j) whereas a ~3-fold increase in lipid droplets >200 μm$^2$ occurred in BAT from Ad-Tmem120a$^{-/-}$ mice on HFD (Fig. 1k, l). Individual organs showed no difference in weight normalised to body weight except a small increase in brain and a shortening of body (~1 cm; 8%) and tail (~0.5 cm; ~5%) length (Supplementary Fig. 2i, j) in Ad-Tmem120a$^{-/-}$ mice on HFD, again specific to females.

**Obesity-resistance in Ad-Tmem120a$^{-/-}$ mice is associated with insulin resistance**. To determine the impact of adipose Tmem120a deletion on metabolic homoeostasis we performed glucose (GTT) and insulin tolerance tests (ITT). Female Ad-Tmem120a$^{-/-}$ mice showed a pronounced impairment of glucose tolerance on HFD compared to control littermates that could still be observed, albeit more mildly, on LFD (Fig. 2a-c), suggestive of insulin resistance. We confirmed this with ITT, where Ad-Tmem120a$^{-/-}$ mice showed relatively impaired glucose suppression response to insulin compared to control littermates on HFD (Fig. 2d). Consistent with prevailing insulin resistance after HFD exposure, fasting serum insulin levels for these mice were markedly higher for Ad-Tmem120a$^{-/-}$ than control littermates (Fig. 2e). Interestingly, regarding sex differences, male Ad-Tmem120a$^{-/-}$ mice showed only moderate differences that were not statistically significant in GTT, but had similarly strong effects as females on ITT (Supplementary Fig. 3d).

Further consistent with the relatively reduced accumulation of fat in individual adipocytes on HFD and adipose–derived insulin resistance, serum leptin (Fig. 2f) and adiponectin (Fig. 2g) levels were reduced in Ad-Tmem120a$^{-/-}$ mice on HFD compared to control littermates. A large study of multiple lipodystrophies reported hyperlipidemia in 42% of all patients; however, for lamin-linked lipodystrophies and metabolic syndromes different studies give widely different frequencies of hyperlididemia, perhaps due to the small numbers of patients[34–36]. Nonetheless, in our mice fasting triglyceride and free fatty acid levels were comparable between genotypes on both diets (Fig. 2h, i and Supplementary Data 1). One possible explanation for the lack of elevated serum triglycerides would be if lipids accumulate in liver; however, there was no difference in liver weight between wild-type and knockout (Supplementary Fig. 2h, i). Liver fat droplet size was also no different and there were no visual indications of hepatic steatosis (Fig. 2j, k). Notably, Ad-Tmem120a$^{-/-}$ mice displayed polydipsia, an excessive thirst observed in diabetes, that is indicative of more severe diabetes on the HFD, compared to littermates (Fig. 2l).

**Ad-Tmem120a$^{-/-}$ mice retain partial carbohydrate utilization on HFD**. To determine whether the relatively reduced accumulation of fat in Ad-Tmem120a$^{-/-}$ female mice on HFD might be related to a change in the carbon substrate used by mitochondria, we performed indirect calorimetry analysis. There were no differences between genotypes on standard diet or after initial exposure to HFD (Fig. 3a–d), but by 8 weeks of HFD divergence in metabolic responses was strong (Fig. 3e, f). Control littermate mice on HFD exhibited the expected suppression of diurnal cycling between fat and carbohydrate utilisation, with a consistent lipid burning respiratory exchange ratio (RER) approaching 0.7 (Fig. 3e). In contrast, Ad-Tmem120a$^{-/-}$ mice maintained a consistently higher RER than littermate controls, indicative of partially retained carbohydrate utilisation across day and night periods (Fig. 3e). Energy expenditure differed significantly between the genotypes (Fig. 3f) despite comparable physical activity (Supplementary Fig. 4a–f).

To assess metabolic activity at the cellular level, we generated adipocytes using isolated primary stromal vascular cells from subcutaneous adipose depots of Tmem120a$^{fl/fl}$ mice. Knockout was achieved in vitro by infection with AAV carrying Cre upon induction of adipogenesis. Notably, knockout cells had smaller lipid droplets upon differentiation compared to the same cells infected with AAV virus carrying GFP only (Supplementary Fig. 4g–i). The differentiated cells were analysed for bioenergetics using a Seahorse flux analyser. Tmem120a knockout cells had higher basal and maximal respiration (Fig. 3g, h and Supplementary Fig. 4j) that was driven by higher proton leak as ATP-linked respiration was comparable to the control. To address the dependency on carbohydrate fuel utilisation, we repeated this analysis with UK5099 to inhibit mitochondrial pyruvate uptake (Fig. 3i, j). Consistent with animals increased carbohydrate utilisation, the knockout adipocytes were more dependent on mitochondrial pyruvate to support respiration. Therefore, whole animal and cellular data both indicate Tmem120a is important in adipocyte fuel substrate utilisation.

**Adipogenic genes are repressed and myogenic genes de-repressed in inguinal fat from Ad-Tmem120a$^{-/-}$ Mice**. As TMEM120A functions in genome organisation/ expression regulation[19,37], we considered that the multiple effects on adipocyte metabolism and function might reflect multiple genes affected by TMEM120A loss. Indeed, RNA-Seq on inguinal subcutaneous WAT stores revealed 116 genes altered in expression in Ad-Tmem120a$^{-/-}$ mice compared to control littermates (Fig. 4a and Supplementary Data 2). To determine if these genes had adipocyte functions that could potentially explain the metabolic defects in the mouse, we extracted their Gene Ontology (GO) terms and separately plotted the fraction of down- or up-regulated genes with a particular (GO) Biological Process. Strikingly, the GO-terms most commonly associated with downregulated genes strongly related to metabolism: out of the total downregulated genes in the knockout >40% had metabolism functions (Fig. 4b). Focusing on GO-terms associated with the most downregulated genes revealed that gene functions reduced more than 10-fold included gluconeogenesis and aromatic amino acid metabolism. Individual gene product links to adipocyte function and metabolism ranged from formyl peptide receptor 1 (Fpr1) that interacts with gut-microbiota generated N-formyl peptide to promote obesity-linked glucose intolerance[38] to protein tyrosine phosphatase receptor type V (Ptprv) that regulates insulin secretion[39] to Nos1ap that increases oxidative stress in diabetes[40] and contributes to adipose tissue beiging[41]. Thus, distinct functions of different genes altered in the knockout could explain different aspects of the lipodystrophy pathology and comorbidities.

The genes most strongly upregulated encoded pro-myogenic proteins or factors that suppress adipogenesis. Plotting the fraction of upregulated genes with a particular GO Biological Process out of the total upregulated genes in the knockout revealed nearly all the top functions relating to muscle (Fig. 4c). Strikingly, most upregulated genes normally are preferentially expressed in muscle (Fig. 4d): thus, it seems Tmem120a strengthens adiposity in part by enhancing repression of genes from this related tissue. This parallels previous findings of a muscle specific NET that represses adipose genes in muscle[18].

**Tmem120a repositions genes between the nuclear interior and the NE during 3T3-L1 adipogenesis**. To globally determine

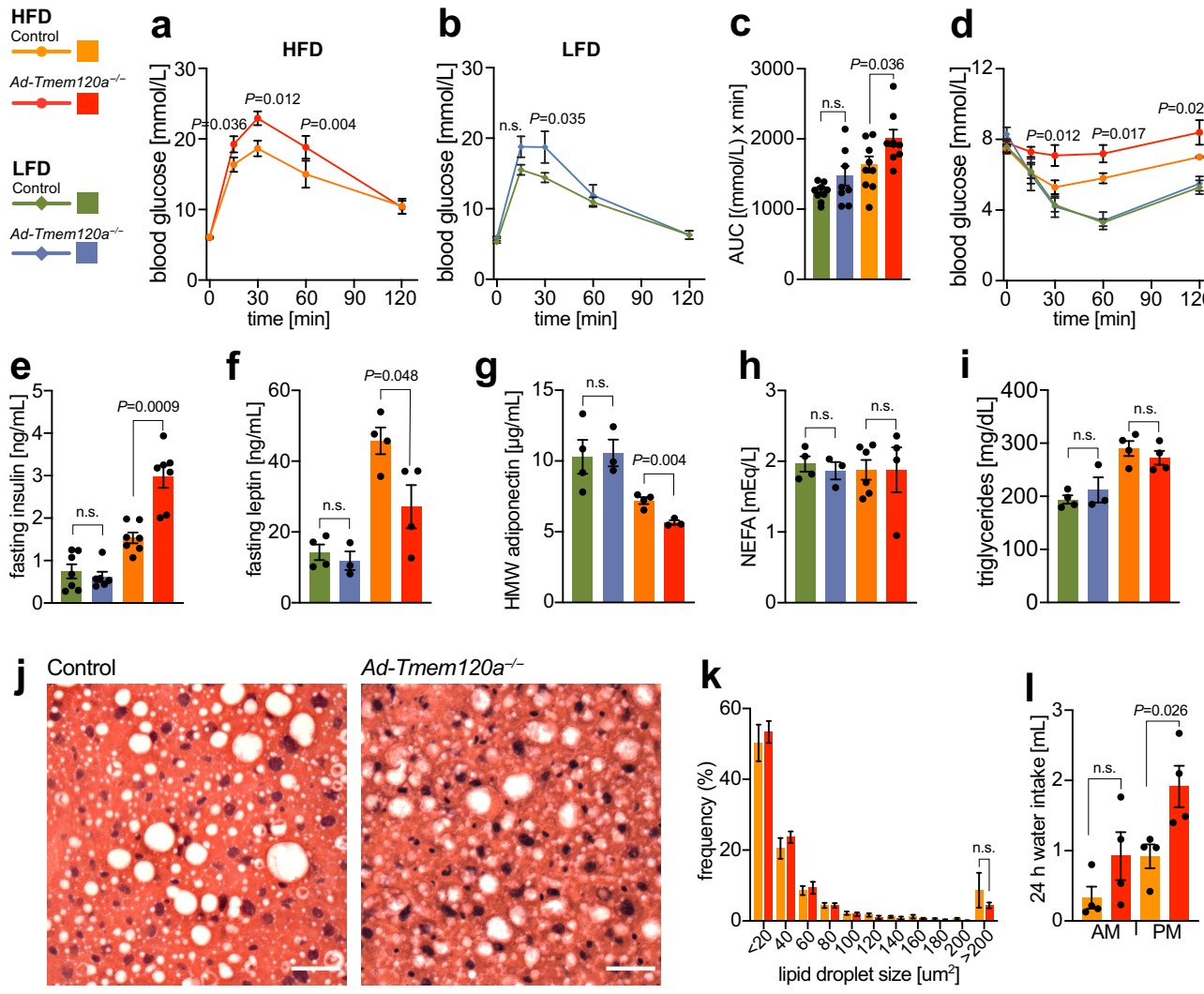

**Fig. 2 Insulin resistance in *Ad-Tmem120a*<sup>-/-</sup> mice.** Glucose tolerance test in control and *Ad-Tmem120a*<sup>−/−</sup> females on HFD (**a**) and LFD (**b**), fasting for 16 h, and quantification of the area under the curve (AUC) (**c**), panels shown as mean ± SEM, control LFD $n = 9$, *Ad-Tmem120a*<sup>−/−</sup> LFD $n = 8$, control HFD $n = 9$, *Ad-Tmem120a*<sup>−/−</sup> $n = 9$. **d** Insulin tolerance test in control and *Ad-Tmem120a*<sup>−/−</sup> females on LFD vs HFD, fasting for 6 h, shown as mean ± SEM, control LFD $n = 7$, *Ad-Tmem120a*<sup>−/−</sup> LFD $n = 6$, control HFD $n = 9$, *Ad-Tmem120a*<sup>−/−</sup> $n = 6$. Blood biochemistry. Fasting serum insulin (**e**), fasting leptin (**f**), high molecular weight (HMW) adiponectin (**g**), NEFA—nonestrified fatty acids (**h**), and triglycerides levels (**i**) in control and *Ad-Tmem120a*<sup>−/−</sup> females on LFD vs HFD, fasting for 16 h, shown as mean ± SEM, control LFD $n = 4$, *Ad-Tmem120a*<sup>−/−</sup> LFD $n = 3$, control HFD $n = 4$, *Ad-Tmem120a*<sup>−/−</sup> $n = 4$. **j** Representative images of haematoxylin and eosin-stained liver sections from control and *Ad-Tmem120a*<sup>−/−</sup> female mice on HFD, scale bars: 40 μm. **k** Quantification of lipid droplet size in liver form control and *Ad-Tmem120a*<sup>−/−</sup> females on HFD, the sizes of the lipid droplets were assigned to size bins and are shown as frequency, area in μm2, shown as mean ± SEM, control HFD $n = 9$, *Ad-Tmem120a*<sup>−/−</sup> $n = 9$. **l** Mean daily water intake by control and *Ad-Tmem120a*<sup>−/−</sup> females after 10 weeks on HFD, shown as mean ± SEM, (for each group $n = 4$), P values: by unpaired, two-sided Student's *t* test. Source data are provided as a Source Data file.

genome regions changing between the NE and interior during adipogenesis and the subset of these under Tmem120a regulation we used DamID that fuses bacterial dam methylase to laminB1 to confer unique methylation to NE-associated DNA so it can be enriched and identified[42,43]. As this approach was not amenable to primary adipocytes we used the well characterized 3T3-L1 in vitro adipogenesis system[44,45] that has the advantages of robust transfectability and differentiation and a wealth of literature to relate positioning data to other genomic datasets. Moreover, it allowed us to test for functional redundancy between Tmem120a and its paralog Tmem120b. LaminB1-DamID profiles were obtained for control 3T3-L1 pre-adipocytes and differentiated adipocytes with or without *Tmem120a* knockdown or *Tmem120a/Tmem120b* knockdown. As Tmem120a is barely detectable in 3T3-L1 pre-adipocytes and strongly induced upon

induction of differentiation, no detrimental effects of expressing knockdown shRNAs prior to differentiation were expected. *Tmem120a* knockdown yielded ~90% reduction in protein levels compared to an integrated empty shRNA vector control in differentiated cells as previously described[21]. Cells were transduced with laminB1-Dam methylase lentivirus either undifferentiated or at 6 days after pharmacological induction of adipogenesis.

Genomic DNA was isolated from pre-adipocytes after 2 days to ensure cells did not reach confluency (that partly induces differentiation) or from 3-day transduced adipocytes (day 9 of differentiation), enriched for Dam methylated DNA, sequenced, and Log2(LaminB1 Dam/soluble Dam) ratios were generated to identify lamina associated domains (LADs). There was little difference in the total percentage of the genome in LADs, being between 43.7% and 45.1% in all conditions (Fig. 5a;

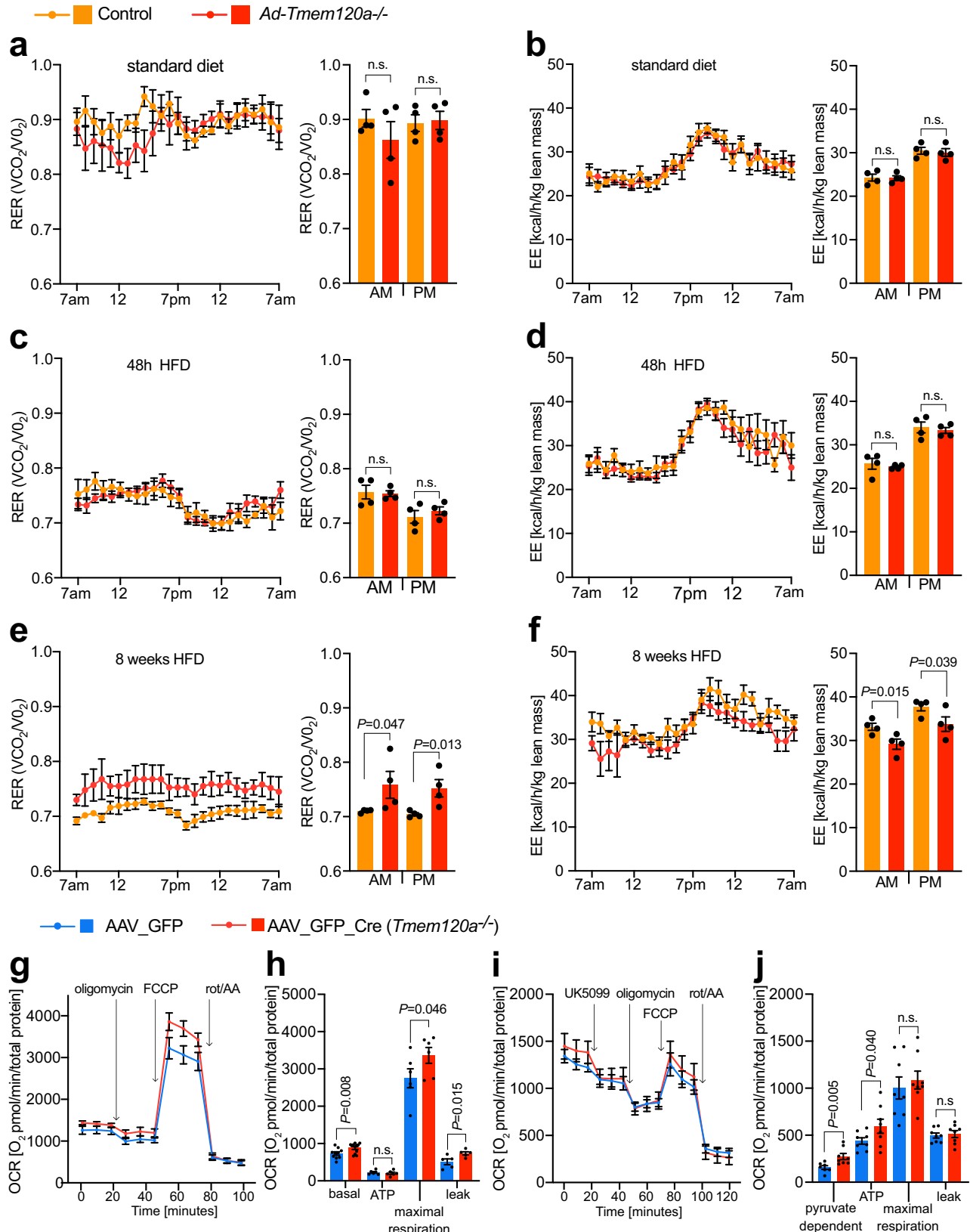

Supplementary Data 3). Similar to DamID in other differentiation systems[18,46,47], the vast majority (~90%) of pre-adipocyte LADs were retained in adipocytes: for those changing, referred to as Differential Regions (DRs), 3.9% of the genome was in LADs that were lost and 4.1% in LADs that were gained during adipocyte differentiation (Fig. 5a and Supplementary Data 3). Similar to a

study on muscle NETs in myogenesis[18], DRs tended to be at the edges of pre-existing LADs that either truncated (87%) or became expanded (53%) upon differentiation.

Of the genes located within adipogenesis DRs, 577 were in newly formed LADs and 1,104 in lost LADs. Roughly a third of those released (385; 35%) were under Tmem120a positional

**Fig. 3 Oxidative metabolism is altered in *Ad-Tmem120a*⁻/⁻ mice.** Indirect calorimetry on standard diet female control and *Ad-Tmem120a*⁻/⁻ mice, respiratory exchange rates—RER (VCO₂/VO₂) (**a**), and energy expenditure—EE [kcal/h/kg lean mass] (**b**). RER trace (**c**) and EE (**d**) in control and *Ad-Tmem120a*⁻/⁻ female mice treated with high fat-diet for 48 h. RER trace (**e**) and EE (**f**) in control and *Ad-Tmem120a*⁻/⁻ female mice treated with high fat-diet for 8 weeks. Line graphs (**a–f**) show representative 24 h trace means and error bars ± SEM and bar graphs show 48 h means and error bars show ± SEM. Indirect calorimetry experiment was performed once and data (EE and RER) from 48 h run were analysed, control HFD *n* = 4, *Ad-Tmem120a*⁻/⁻ *n* = 4. **g**, **h** Oxygen consumption rates (OCR) in stromal vascular fraction cells (SVF) differentiated into adipocytes isolated from lox/lox animals infected with virus AAV_GFP_Cre to obtain *Tmem120a* knockout and as control SVF cells infected with GFP virus AAV_GFP. Mitochondria stress experiment in the presence of oligomycin, an ATPase inhibitor, FCCP—mitochondrial respiration uncoupler (maximal respiration) and rotenone with antimycin A (rot/AA)—electron transport chain complex I and III blockers, respectively. OCR traces normalised to total protein are shown as means SEM (**g**) and non-mitochondrial respiration-normalised quantification of OCR is shown on a bar chart graph (**h**) as means SEM, (*n* = 6 separate AAV infection per group). **i**, **j** Pyruvate dependency in SVF adipocytes from lox/lox mice subcutaneous fat infected with AAV_GFP or AAV_GFP_Cre as OCR was measured upon delivery of UK5099. *P* values: by unpaired, two-sided Student's *t* test. See also Supplementary Fig. 4. Source data are provided as a Source Data file.

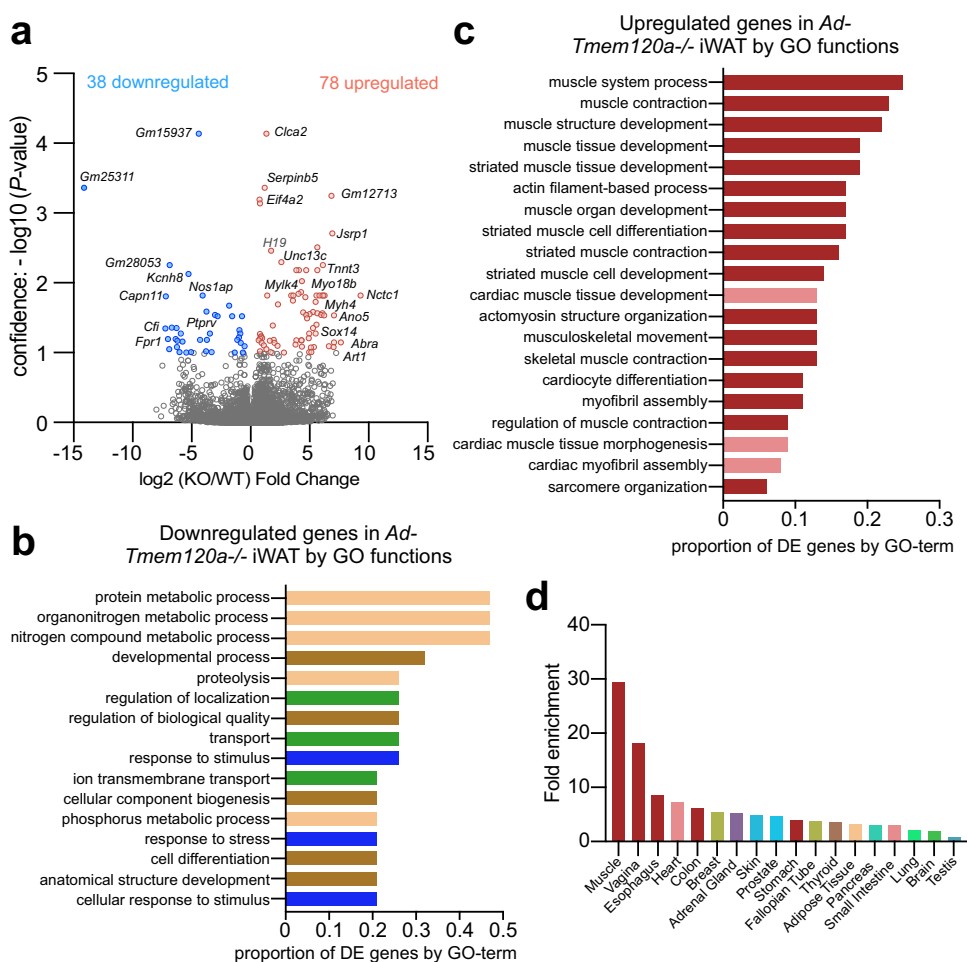

**Fig. 4 *Ad-Tmem120a*⁻/⁻ mouse exhibits de-repression in inguinal WAT of many genes that normally repress myogenesis. a** Volcano plot of genes altered in inguinal subcutaneous white adipose tissue from control and *Ad-Tmem120a*⁻/⁻ female mice on HFD (*n* = 5). **b** Gene Ontology (GO) Biological Process analysis for downregulated genes. GO terms found to be significantly enriched using g:Profiler fall into four classes: metabolism (beige), development (brown), transport (green) and response to stimulus (blue). The bars represent the proportion of genes associated with each individual enriched GO term that are present in the downregulated genes dataset. **c** Gene ontology (GO) Biological Process of analysis for upregulated genes. All of the enriched GO terms are related to muscle function, either skeletal (dark red) or cardiac (pale red). The bars represent the proportion of genes associated with each individual enriched GO term that are present in the upregulated genes dataset. **d** Tissue-specific gene enrichment for upregulated genes in iWAT from *Ad-Tmem120a*⁻/⁻ mouse by TissueEnrich tool presented as fold enrichment. Source data are provided as a Source Data file.

regulation, failing to be released when *Tmem120a* was knocked down. The numbers increased to 43% (480 genes) when considering those that failed to be released in the *Tmem120a/Tmem120b* double knockdown. For those in newly formed adipocyte LADs, 24% (136) were under Tmem120a

positional regulation and 28% (161) were under combined Tmem120a/Tmem120b positional regulation.

To directly match repositioning and expression changes, we used microarrays on RNA isolated from parallel samples to the DamID. GO analysis for these revealed similar functions as the

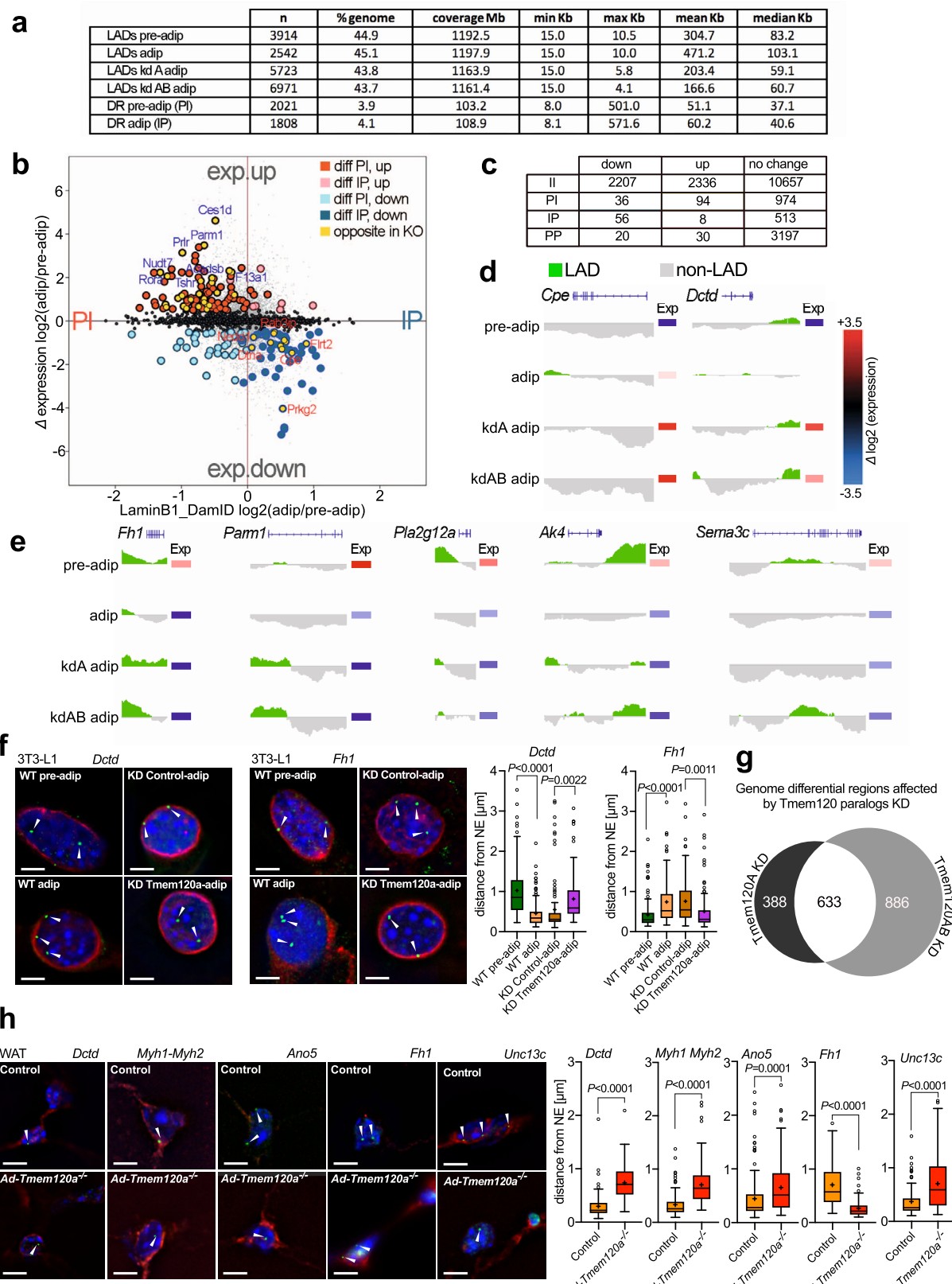

genes changing expression in iWAT from the _Tmem120a−/−_ mice (Supplementary Fig. 5a, b). Intersecting the set of genes changing position and those changing expression during differentiation revealed that ~10% of genes that changed position also changed expression, also consistent with previous studies. Of the 1,104 genes in lost LADs, 94 increased expression while, of the

577 genes recruited to newly formed LADs, 56 reduced expression (Fig. 5b, c; Supplementary Data 3). A subset of these were under Tmem120A positional regulation (highlighted in yellow), but they tended to be relevant to fat metabolism and lipodystrophy associated comorbidities. For example, genes at DRs normally moving to the NE to become repressed that failed

**Fig. 5 DamID and transcriptomics analysis reveals Tmem120a contributes to gene regulation in adipocyte differentiation. a** LaminB1-DamID on 3T3-L1 preadipocytes and adipocytes, upon *Tmem120a* knockdown as well as *Tmem120a* and *b* double knockdown. Table shows metrics for lamina associated domains (LADs). **b** Correlation of gene expression and gene positioning shown as distribution scatter plot of genes differentially expressed (exp.up/exp.down) during adipogenesis (adip/pre-adip) and LamB1-DamID values, PI—genes moving from periphery to nuclear interion, IP—genes moving toward nuclear periphery. **c** Table showing numbers of genes changing expression upon being repositioned between nuclear interior "I" and nuclear periphery "P" during adipogenesis. Example DamID traces and expression changes of two genes being recruited to the NE (IP) upon differentiation that failed to reposition with *Tmem120a* knockdown (**d**), and examples for the opposite direction (PI) (**e**). **f** FISH in 3T3_L1 system for the *Dctd* and *Fh1* genes in green with white arrow heads during wild-type adipogenesis and in the *Tmem120a* knockdown, distance from NE of n = 105 loci per group and gene, Tukey box plot representing median, cross on the box represents mean, bounds of box represents interquartile of the data, whiskers representing minima/maxima excluding outliers and dots represents outliers of more than 2/3 times of upper quartile, bars: 5 μm. **g** Venn diagram showing the number of genomic DRs affected by Tmem120 paralogs, representing the redundancy effect of single (Tmem120a) and double (Tmem120ab) knockdown on the genome organisation. **h** FISH in mouse iWAT from control animals vs. *Ad-Tmem120a*$^{-/-}$ detecting the position of the *Dctd, Myh1/Myh2, Ano5, Fh1* and *Unc13c* genes; scale bars: 5 μm; distance from NE quantification is shown as Tukey box plot representing median, cross on the box represents mean, *Dctd* (WT n = 89 loci and KO n = 96 loci), *Myh-1,2* (WT n = 105 loci, KO n = 105 loci), *Ano5* (WT n = 165 loci, KO n = 138 loci), *Fh1* (WT n = 111 loci, KO n = 83 loci), Unc13c (WT n = 90, KO n = 110). *P* values were calculated by unpaired two-sided, Student's *t* test. See also Supplementary Fig. 5. Source data are provided as a Source Data file.

to move and failed to be repressed in the Tmem120a knockdown included *Cpe* that yields obesity in mice when defective[48] and *Dctd* that has altered genome interactions in fat morbidities[49] and increased SNPs in childhood obesity[50] (Fig. 5d). Notably, the *Cpe* gene was directly next to rather than overlapping with a newly formed LAD in adipogenesis, suggesting that promoters can be specifically targeted by this regulation.

Proteins encoded by DR genes normally released from the NE and activated in adipogenesis that failed to do so in the Tmem120a knockdown include Fumarate hydratase (Fh1) that is important for respiration[51], Parm1 that is associated with insulin resistance[52], Jazf1 that is associated with type 2 diabetes[53], lipid metabolism proteins Pla2g12a and Ak4[54–56], and adipokine Sema3c[57] (Fig. 5e). Expression of these genes increased in differentiated adipocytes, but this was mitigated with *Tmem120a* knockdown. Thus, multiple genes involved in different aspects of fat metabolism were reduced upon loss of Tmem120a. The repositioning of both *Dctd* and *Fh1* was confirmed by FISH (Fig. 5f and Supplementary Fig. 5c).

Notably, it could be observed in some of the DamID traces (Fig. 5d, e) that there were differences between the *Tmem120a* knockdown and the *Tmem120a/Tmem120b* double knockdown. For *Parm1* and *Pla2g12a* the traces were very similar between the two, suggesting that the change was specific to loss of the Tmem120a protein and Tmem120b had no effect. In contrast, for *Cpe* and *Fh1* the change in the LAD intensified in the double knockdown compared to the *Tmem120a* knockdown alone. For *Dctd* and *Ak4* there were additional LADs added in the double knockdown that did not occur in the *Tmem120a* knockdown alone. Finally, for *Sema3c* the *Tmem120a* knockdown had no effect at the locus, but the LAD was uniquely contributed by the Tmem120b because the change only occurred at all in the double knockdown. This led us to question the extent to which the targets of Tmem120a and Tmem120b might differ; so we set parameters to search for LADs that appeared equally in the *Tmem120a* knockdown and the double knockdown sample versus LADs that were unique to the *Tmem120a* knockdown sample or to the double knockdown sample. This revealed that roughly 1/3 (633) of the affected loci were uniquely affected only by Tmem120a with no further change at the locus when both proteins were knocked down while those only affected by Tmem120b (unique to the double knockdown sample) were close to 50% (886) (Fig. 5g). The smallest group were loci for which Tmem120a and Tmem120b seemingly had different genes within each locus (~20% [388] of the affected loci Fig. 5g). Thus, there is strong evidence for considerable functional diversity between Tmem120a and Tmem120b targets.

***Ad-Tmem120a*$^{-/-}$ WAT shows impairment of Tmem120a regulated gene–repositioning observed in 3T3-L1 adipogenesis.** We next tested if *Ad-Tmem120a*$^{-/-}$ mice recapitulated gene repositioning changes of the 3T3-L1 system. FISH was performed on paraffin-embedded and sectioned iWAT from wild-type or knockout animals. All genes tested confirmed the Tmem120a dependence of repositioning. Anti-adipogenic *Dctd*, that repositioned to a LAD during 3T3-L1 adipogenesis, failed to be recruited to the NE in iWAT from *Ad-Tmem120a*$^{-/-}$ mice (Fig. 5h). Moving in the opposite direction, *Fh1* that is normally released from the NE during 3T3-L1 adipogenesis, was away from the NE in control littermate iWAT; however, in iWAT from the *Ad-Tmem120a*$^{-/-}$ mice this gene was at the NE as in the undifferentiated 3T3-L1 pre-adipocytes (Fig. 5h).

As genes associated with myogenesis were also highlighted by both DamID and expression data, we similarly tested genes encoding muscle myosin heavy chain 1 and 2 (*Myh1* and *Myh2*), calcium-activated chloride channel Ano5 that is mutated in limb-girdle muscular dystrophy[58], Myo18b, and Unc13c that are all recruited to the NE and repressed in 3T3-L1 adipogenesis (Supplementary Fig. 5d and Supplementary Data 3 and 4). The repositioning of *Myo18b*, *Myh1*, and *Myh2* was confirmed by FISH (Supplementary Fig. 5e). This was paralleled in the mouse system with the genes in the nuclear interior in *Ad-Tmem120a*$^{-/-}$ mice, indicating that they had failed to be recruited to or maintained at the NE (Fig. 5h and Supplementary Fig. 5f). All tested genes also in parallel exhibited corresponding expression changes in the mice (Supplementary Data 3 and 4). Importantly, these repositioning changes in the mice occurred not only on the HFD but also on the LFD (Supplementary Fig. 5g), indicating that the gene repositioning occurs as a result of the knockout and not as a result of the change in diet. Thus, the phenotypic change must reflect a defect in the ability of the knockout cells to respond to the challenge of the metabolic load of the HFD, due to the many metabolic and other genes altered in the knockout.

**Enhancers and regulatory RNAs were also altered in *Ad-Tmem120a*$^{-/-}$ mice.** While the above genes changed both expression and position in the mice, some genes with changed expression in the knockout did not correspondingly change position. We previously found that many enhancers are recruited to or released from LADs during lymphocyte activation[59]. Therefore, we looked for predicted enhancers (source: Enhancer Atlas http://www.enhanceratlas.org) within genomic regions that are recruited to or released from the NE (i.e.: DR regions) during 3T3L1 differentiation. A third of the enhancers that are normally

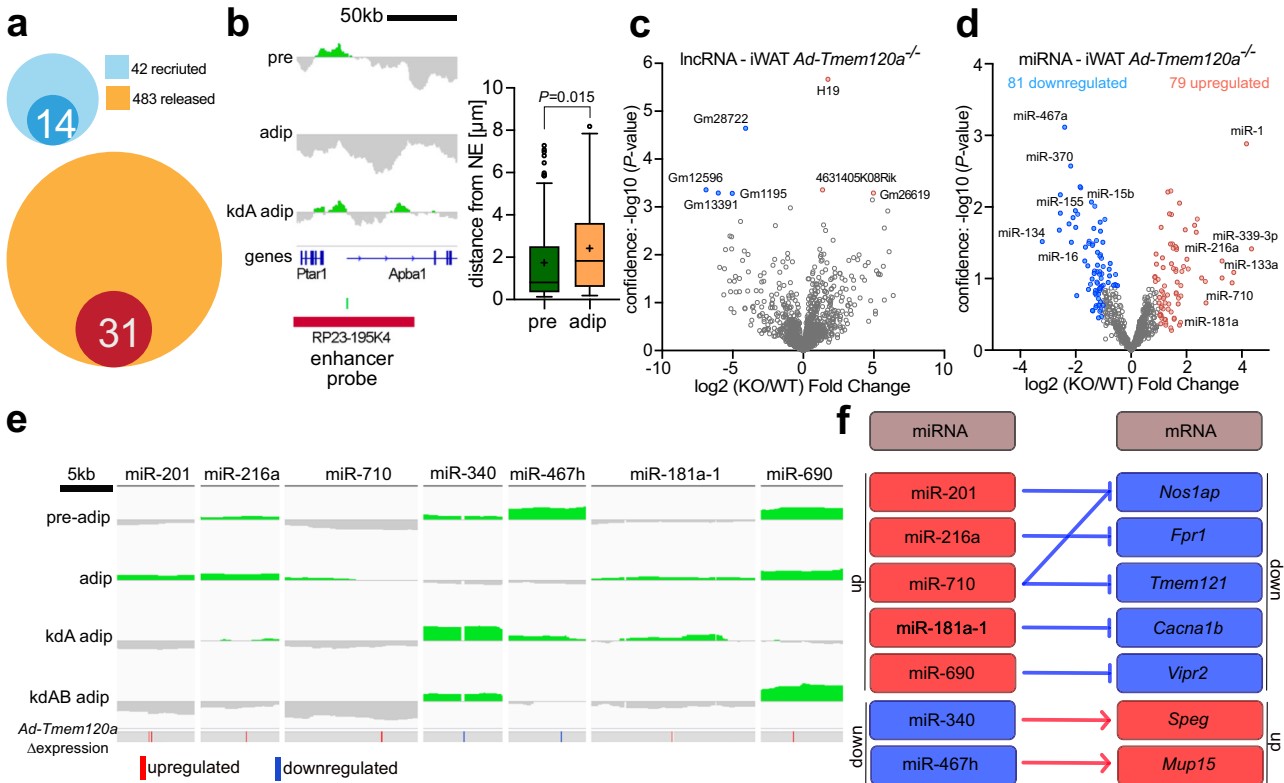

**Fig. 6 Enhancers and regulatory RNAs under NE positional control. a** Venn diagrams showing proportion of predicted adipogenic enhancers that become lost during adipogenesis—recruited to NE (light blue) or new adipocyte enhancers – released from NE (yellow). Dark blue circle represents enhancers that fail to recruit to NE during adipogenesis in Tmem120a KD 3T3-L1 cells. Red circle represents enhancers that fail to release form NE during Tmem120a KD 3T3-L1 differentiation. **b** DamID trace and FISH loci signal distance from NE quantification of the Klf9 enhancer shown as Tukey box plot representing median, cross on the box represents mean, bounds of box represents interquartile of the data, whiskers representing minima/maxima excluding outliers and dots represents outliers of more than 2/3 times of upper quartile. Pre-adipocytes (pre) and adipocytes (adip) $n = 73$ loci for both groups. P value was calculated by unpaired, two-sided, Student's $t$ test, **c** Volcano plot showing expression of lncRNA, highlighting strong upregulation of the H19 expression in *Ad-Tmem120a$^{-/-}$* iWAT. **d** Volcano plot of differentially expressed miRNAs in iWAT. **e** DamID traces showing the position of miRNA-encoding loci with respect to changing LADs and their expression change in the knockout mouse. Blue, downregulated; red, upregulated. **f** Target genes of differentially expressed miRNAs show corresponding expression changes (RNA-Seq). See also Supplementary Data 5. Source data are provided as a Source Data file.

recruited to the NE failed to do so when Tmem120A was knocked down, while 31 enhancers that should be released from the NE remained at the NE upon Tmem120a knock-down (Fig. 6a). We tested the positioning of one of these predicted to act on the *Klf9* gene that is in a pre-adipocyte LAD that disappears upon differentiation, yet is retained in adipocytes knocked down for Tmem120a. FISH confirmed this predicted enhancer releases from the NE during adipogenesis (Fig. 6b).

Another way to explain Tmem120a effects on gene expression absent positional changes in the affected genes is changes in regulatory RNAs. Indeed, RNA-Seq on the mouse iWAT indicated changes in lncRNA levels (Supplementary Data 2), with H19, a lncRNA involved in skeletal muscle and heart differentiation[60] upregulated 3.2-fold in *Ad-Tmem120a$^{-/-}$* mice compared to control littermates (Fig. 6c). This raised the possibility of other indirect effects on gene expression from changes to other regulatory RNAs.

Levels of 717 miRNAs were determined in control and knockout mouse iWAT by high-throughput qPCR (Fig. 6d and Supplementary Data 5). Of 79 upregulated miRNAs, 45 had relevant links to the *Ad-Tmem120a$^{-/-}$* pathology. Links were found for 19 (24.1%) to myogenesis, 9 (11.4%) to adipogenesis, and 25 (31.6%) to insulin resistance or obesity (Supplementary Data 5, sheets 1,2). miR-339-3p, the most upregulated, was increased over 20-fold and affects Akt signalling pathways important for insulin regulation[61,62]. The next most upregulated

were miR-1 at 18-fold and miR-133a at 9.8-fold that work together to promote muscle differentiation[63]; hence upregulation of these miRNAs may enhance de-repression of myogenic pathways. They are also linked to diabetes and miR-133a is additionally involved in WAT browning and so could explain BAT fat accumulation in *Ad-Tmem120a$^{-/-}$* mice.

In the opposite direction, of 81 miRNAs with decreased levels, 60 had links to *Ad-Tmem120a$^{-/-}$* pathology. These connections included 10 miRNAs (12.3%) related to myogenesis, 30 (37%) to adipogenesis, and 37 (45.7%) to insulin resistance or obesity (Supplementary Data 5). The strongest decrease was 9.2-fold and the average fold change was 3.3-fold. Of particular note, knockdown of miR-27b, miR-92a, miR-155, and miR-150 have all been shown to enhance browning/ beiging[64–67], one of the *Ad-Tmem120a$^{-/-}$* pathologies. Also, miR-322, miR-15b, miR-342-5p, and miR-16 are all pro-adipogenic[68–70]; so their reduction is consistent with the pathology. Finally, miR-151-3p, miR-466f, miR-139-5p, miR-219, miR-29b, miR-25, and miR-370 are all linked to diabetes[71–77], consistent with the inability to clear glucose, insulin resistance, and elevated insulin in *Ad-Tmem120a$^{-/-}$* mice.

Of these 160 mis-expressed miRNAs, 67% were found in serum from ADicerKO mouse, a lipodystrophic mouse, and 9 were misregulated in serum from human lipodystrophy patients[78] (Supplementary Data 5, sheet 1). Moreover, 33 of the loci encoding them were under positional regulation according to the

DamID data (Supplementary Data 5, sheet 1). Surprisingly, a majority of those under Tmem120a positional regulation did not change position during normal adipocyte differentiation, but all of those normally repositioning in adipogenesis failed to reposition in *Tmem120a* knockdown cells. Examples include the loci encoding miR-201, miR-216a, and miR-710 that increase at the NE during adipogenesis and fail to do so in *Tmem120a* and *Tmem120/Tmem120b* knockdowns, corresponding to their increased expression (Fig. 6e). miR-201 may contribute to fatty acid metabolism as it targets very-long-chain enoyl-CoA reductase[79] while miR-216a is a risk biomarker for atherosclerosis in paediatric patients with diabetes[80]. In the opposite direction, the loci encoding miR-340 and miR-467 were released from the NE in adipogenesis and failed to release with *Tmem120* knockdown, corresponding with their reduced levels (Fig. 6e). miR-340 is linked to gestational obesity[81] and type 2 diabetes[82]. The locus encoding miR-181a exhibited an intermediate effect. It was recruited to the NE during normal adipogenesis, but, when *Tmem120a* was knocked down, the LAD it occurs in was shortened. In the double knockdown the LAD completely disappeared. In the mice expression of the miRNA locus was increased even though *Tmem120a* alone was knocked out. miR181a appears to be important for adipogenesis in both porcine[83] and 3T3-L1 mouse models[84] and interestingly, considering Tmem120a effects on circadian switching for oxidative metabolism, it modulates circadian rhythm in adipose-derived stromal cells[85]. Some repositioning changes were harder to interpret. For example, miR-690 remained in a LAD during normal adipogenesis yet was released from the NE with *Tmem120a* knockdown but not the double knockdown (Fig. 6e). miR-690 regulates induced pluripotent stem cell differentiation into insulin-producing cells[86] and is downregulated in high glucose conditions[87]. In all these cases, the relevant target genes of the miRNAs exhibited corresponding changes in our knockout mouse RNA-Seq data (Fig. 6f). This was not only the case for the specific miRNAs shown, but for the wider set found to change in the knockout (Supplementary Data 5, sheet 3).

**Gene mispositioning of *Ad-Tmem120a*$^{-/-}$ mice is recapitulated in human partial lipodystrophy patients.** As Tmem120a is in the NE and *Ad-Tmem120a*$^{-/-}$ mice had a latent lipodystrophy pathology, we postulated that TMEM120A might play a mediating role in FPLD2, caused by mutations in *LMNA*. If so, the gene mispositioning we observed in *Ad-Tmem120a*$^{-/-}$ mice might also occur in human FPLD2 patients. Therefore, human FISH probes were used to test for positioning of 4 equivalent loci in human FPLD2 patient cells. Pre-adipocytes were isolated from neck fat biopsies from two FPLD2 patients and a healthy control and differentiated in vitro. As the strongest gene expression changes in the mouse were for myogenic genes, we chose 5 muscle genes that normally reposition from the nuclear interior to the NE in mouse adipogenesis and failed to reposition in the *Ad-Tmem120a*$^{-/-}$ mice corresponding to their derepression in mouse iWAT: *MYH1*, *MYH2*, *ANO5*, *TRIM55*, and *MYO18B*. We first tested these for repositioning in human adipogenesis as there is commonly incomplete correspondence between mouse and human in genome organisation studies due to synteny differences and the fact that sometimes a different variant of the same protein family is used for a particular function between the two species. Because the *MYH1* and *MYH2* genes are adjacent to one another a single BAC probe was used to cover both genes. *MYH1*, *MYH2*, *ANO5*, *TRIM55* all repositioned as expected, but *MYO18B* was positioned internally in both the pre-adipocytes and adipocytes in the control and so did not reposition and was not followed further (Fig. 7a-d). When the remaining genes that all similarly repositioned between the nuclear interior and NE during adipogenesis in both mouse and human were tested in the FPLD2 patient cells, all four genes failed to reposition in the lipodystrophy patients upon adipogenic differentiation while they did in the controls (Fig. 7a-c). Since Tmem120a protein levels were unchanged in FPLD2 patients (Supplementary Fig. 6), we suggest FPLD2-mutated A-type lamins directly or indirectly affect the localization and/or function of Tmem120.

## Discussion

*Tmem120a* disruption in mice yields a latent lipodystrophy pathology resembling in many ways *LMNA*-linked FPLD2, including a pronounced gender dimorphism of currently unknown molecular etiology. The correspondence between the mispositioning of several specific genes, enhancers and miRNAs supporting adipocyte differentiation with concomitant changes in expression, metabolism, and mouse pathology suggests that the latent lipodystrophy pathology arises due to disruption of fat-specific patterns of genome organisation and the inability of the cells with these multiple alterations in adipocyte gene expression to handle the increased metabolic loads on the HFD. That similar genome organisation defects occurred in human FPLD2 caused by *LMNA* mutation raises the possibility that TMEM120A might mediate the fat-specificity of this disorder as A-type lamins are widely expressed and the lamina polymer they form under the inner nuclear membrane interacts with many nuclear membrane proteins. Indeed, it was previously published that Tmem120a is in the inner nuclear membrane and resists a pre-fixation detergent extraction that is characteristic of an interaction with the lamin polymer[20], and we have recently found that lamin A can be pulled down with Tmem120a and the two purified proteins expressed in bacteria can interact (unpublished data). The potential of TMEM120A to contribute to human adiposity disorders is further increased by recent findings of point mutations in muscle-specific NETs with parallel genome-organizing functions in Emery-Dreyfuss muscular dystrophy[88] that similarly has variants caused by *LMNA* mutations. Thus, these findings have ramifications for spatial genome organisation, fat development and metabolism, and human disease.

The latent lipodystrophy pathology of *Ad-Tmem120a*$^{-/-}$ mice is more similar in some ways to human FPLD2 than the lamin A FPLD2 mouse model expressing a human lamin A R482Q transgene from the aP2 promoter[89]. Both our *Ad-Tmem120a*$^{-/-}$ mice and the aP2-R482Q mice exhibited insulin resistance, glucose intolerance, and fat loss with a depot-specific pattern consistent with human FPLD2. The aP2-R482Q mice more closely parallel human FPLD2 in having hyperlipidemia and hepatic steatosis due to accumulation of lipids in non-adipose tissues. In contrast, *Ad-Tmem120a*$^{-/-}$ mice had no hyperlipidemia, no increase in hepatic lipid droplet size, and no significant increase in liver mass. This may indicate a distinct Tmem120a-linked lipodystrophy pathology with translational significance given a recent report that hepatic steatosis occurs in only ~10% of lipodystrophy patients[34] and that hyperlipidemia is variable in the human disorder[36]. The implication for the latter argument is that lamin A mutation effects on liver are largely direct, as opposed to indirect from lipids that cannot be processed by fat stores. Intriguingly, our model may eventually illuminate mechanisms underpinning sex-specificity of lipodystrophy as Ad-Tmem120a knockout mice had more pronounced effects in females while the lamin A R482Q mutant line yielded pathologies, including the hepatic steatosis, principally in males[90]. Furthermore, in FPLD2 loss of subcutaneous fat from the extremities, trunk, and gluteal region is partly countered by increased fat in the neck (where adult human BAT concentrates) and increased muscle mass. The lamin A R482Q model had a severe reduction in brown

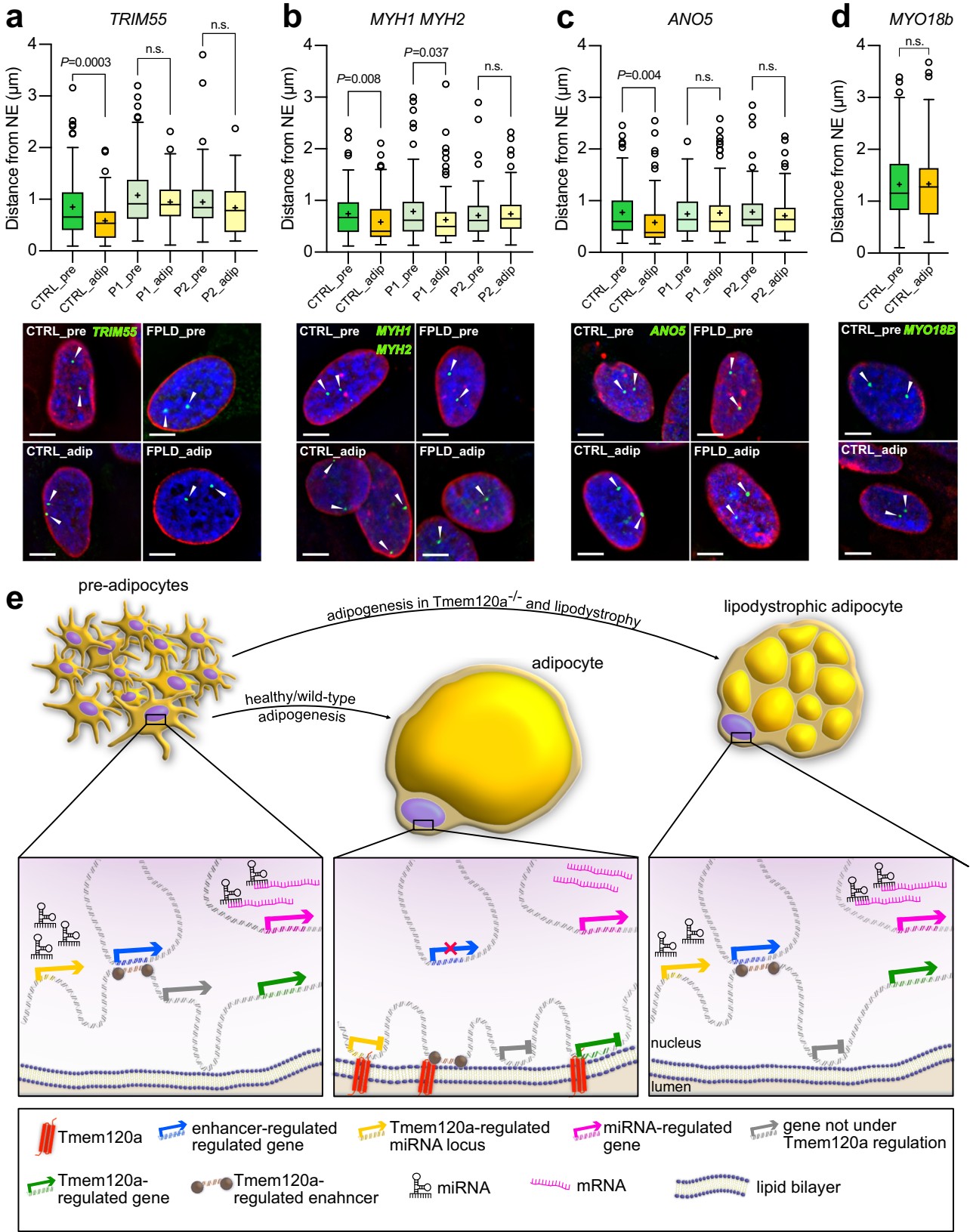

subscapular fat pads and no significant increase in lean mass. In this, our adipose tissue specific *Tmem120a* knockout mice were closer to the human disorder having both maintained BAT mass and increased skeletal muscle mass.

We speculate that the increased fat accumulation in *Ad-Tmem120a*$^{-/-}$ BAT parallels the fat accumulation in the necks of FPLD2 patients[91,92], especially as adipose tissue browning is observed in FPLD2[93]. BAT stores less lipid, has more mitochondria and can dissipate energy as heat without ATP generation whereas WAT is less efficient at converting lipid to energy but more efficient at lipid storage[94,95]. Thus, the accumulation of fat in BAT suggests a deficiency in BAT metabolic function that

**Fig. 7 Genes misregulated in *Ad-Tmem120a⁻/⁻* mice are also misregulated in human lipodystrophy patients. a–d** Quantification of genetic loci distance from NE shown as Tukey box plot representing median, cross on the box represents mean, bounds of box represents interquartile of the data, whiskers representing minima/maxima excluding outliers and dots represents outliers of more than 2/3 times of upper quartile. $n = 105$ loci per condition/gene/participant, $P$ values were calculated by unpaired two-sided, Student's $t$ test and representative images of FISH for *TRIM55*, *MYH1/MYH2*, *ANO5*, and *MYO18b* gene loci in human primary SVF cells before and after induction of adipogenesis, scale bars: 5 µm: CTRL, healthy donor; pre, preadipocytes; adip, adipocytes; P1 and P2 stands for patient 1 and patient 2 respectively. Cells are from anonymised FPLD patients donating tissue. **e** Model of genome organisation role in adipogenesis and lipodystrophy. In preadipocytes, some muscle genes (e.g., *Myh1*) are in the nuclear interior that need to be strongly shut off in adipocytes and so are recruited to the NE when Tmem120a is expressed during adipogenesis. In the opposite direction, some genes are released from the NE and become activated when Tmem120a is expressed; however, in the absence of Tmem120a they remain at the NE and their expression is reduced compared to wild-type adipocytes. Enhancers and miRNA-encoding loci are similarly under Tmem120a positional and expression regulation. *Ad-Tmem120a⁻/⁻* adipocytes parallel human FPLD2 patients in that the normal repositioning of several genes that takes place during adipogenesis is disrupted with a concomitant change in expression. Source data are provided as a Source Data file.

could be accounted for by both the repressed genes and altered miRNAs. Furthermore, several of the altered miRNAs have reported functions in beiging[64–67] and this could also explain the change in BAT function.

We posit that the increased lean/muscle mass in our model and in human FPLD2[91,92] could be accounted for by the known functions on muscle of several of the miRNAs elevated in our *Ad-Tmem120a⁻/⁻* mice such as miR-1 and miR-133a that work together to promote muscle differentiation[63]. Many of the pro-myogenic miRNAs affected by Tmem120a deficiency were found in serum of human patients with various fat disorders[78] suggesting a potential adipocyte-muscle paracrine mechanism driving increased muscle mass.

That miRNA levels, fat enhancers, and relevant gene expression correlated with genome regions moving between the NE and nuclear interior suggests that the mechanism behind our mouse pathology is disruption of fat-specific genome organisation. The notion that gene positioning influences gene regulation has been criticised because only a fraction of all the genes that change position also change expression and of this fraction only ~70% change expression in the expected direction[18,46]. The favoured explanation for these poor correlations is that multiple adjacent genes reposition together, but only those with transcriptional regulators present change expression[18]. Our data can better explain those changing expression in the unexpected direction through enhancers and regulatory RNA-encoding genome regions that are also under Tmem120a positional control (Fig. 7e). This argument is even stronger when considering that >75% of the altered miRNAs had functions influencing fat or muscle.

A recent report argued that *Tmem120a* encodes a brain ion channel named TACAN[26]. It is possible that the much higher molecular weight isoform in brain we previously reported[21] has a distinct function. However, it is also possible that a genome organisation defect could largely explain their results as ion channels Cacna1s, Clca2, Slco1a5, Ano5, Tmem150c, ATP2a1, Kcnv2, Fxyd2, Slc4a8, and Kcnh8 were all misregulated in our model. Notably, Atp2a1 is a Ca2+ ATPase pump that regulates glucose transport and is linked to diabetes[96,97], further linking Tmem120a indirectly to ion transport and directly to adipocyte function. Our data is further supported by 5 different groups finding human and mouse TMEM120A links to adiposity[21–25].

Cumulatively, these data suggest a paradigm shift for genome organisation defects in disease where multiple minor changes in expression can yield a disease phenotype as opposed to blocking of one central pathway. A few genes from different pathways were reduced ~10-fold while many genes were reduced only a few fold in our knockout, yet it yielded a latent lipodystrophy pathology without any major adiposity genes/pathways being strongly shut down. Such minor changes in multiple genes would be likely to reduce overall efficiency of metabolic function, consistent with

the later presentation of NE-linked lipodystrophies long after tissue differentiation and when metabolic loads are increased[98]. Obesity and other metabolic disorders are increasing throughout the world with age-standardized diabetes in men increasing from 4.3% in 1980 to 9% in 2014[99]. Our findings highlight TMEM120A as a potential target/model for these disorders due to the large range of fat metabolic genes it affects and our finding of minimal phenotype in animals on the low-fat diet supports the clinically favoured approach of caloric restriction to improve metabolic parameters in lipodystrophy[100]. Further work dissecting out TMEM120A critical miRNA targets could potentially enable miRNA therapies targeting separately its muscle mass function for muscle disorders and its fat metabolic functions for obesity.

## Methods

**Animals**. A conditional *Tmem120a* knockout mouse was generated by Taconic. The strategy was targeted to the NCBI transcript NM_172541.2. A targeting vector was generated using BAC clones from the C57BL/6 J RPCIB-731 BAC library and was transfected into the Taconic C57BL/6 N Tac ES cell line. The targeting vector consisted of exons 2 and 3 region flanked by *loxP* sites as well as the puromycin selection cassette designed to be removed after in vivo Flp-mediated recombination (Supplementary Fig. 1a). Recombination of the targeting vector was confirmed by Southern blotting (Supplementary Fig. 1b). Conditional knockout homozygotes with strain background C57BL/6 N Tac (*loxP/loxP*) were crossed with an Adipoq-Cre deletor mouse (background C57BL/6 J) carrying promoter for adiponectin followed by *Cre* recombinase transgene, kind gift from professor Evan Rosen (RRID_IMSR_JAX:028020), described in[31]. Adipoq-Cre deletor mouse was chosen to obtain expression of Cre recombinase in mature adipocytes and to avoid expression in brain, macrophages and muscles observed in other models of adipose tissue specific Cre systems[32,33]. Genotyping for *loxP* sites was performed by PCR with primers flanking two *loxP* sites, F1 5′ CTTTCAGGCTGGGTAGGTAGG, R1 5′ GGGCGTCTGAATTCTAAGTGG and F2 5′ GAGTTCAAAGTTAGCC TAGTCAGC, R2 5′ TCCAGAAGGTGGTATCAGATCC. To genotype Adipoq-Cre mice, detection of the transgene was performed by PCR and use of the following pairs of primers: *Cre* forward 5′ GCGGTCTGGCAGTAAAAACTATC, reverse 5′ GTGAAACAGCATTGCTGTCACTT, in combination with the internal positive control (*Il2* gene): forward 5′ CTAGGCCACAGAATTGAAAGATCT, reverse 5′ GTAGGTGGAAATTCTAGCATCATCC. Animals were kept in an air-conditioned room with ambient temperature of ~20 °C and humidity between 40 and 60% in 12 h light and dark cycles (light on 7 am) with *ad libitum* access to water and food. Experiments involving animals were approved by the University of Edinburgh ethical committee and Home Office UK under project licenses granted to ECS and NMM.

**In vitro adipocyte differentiation and stable shRNA Lines**. 3T3-L1 (RRID_0123) cells were maintained in high glucose DMEM (SIGMA, D5796) supplemented with 10% foetal bovine serum (FBS, Gibco, 10270), MEM non-essential amino acids, 1 mM Sodium Pyruvate (Gibco, 11360-039), 100 µg/µl penicillin and 100 µg/µl streptomycin sulphate (Gibco, 15140-122). Cells were plated at ~10% confluency and always split by 60% confluency to maintain differentiation capacity. Adipogenesis was induced two days post-confluency by replacing growth medium with induction medium containing 0.5 µM insulin (SIGMA, I1882), 1 µM dexamethasone (SIGMA, 04902), 0.5 mM 3-isobutyl-1-methylxanthine (IBMX, SIGMA, I7018), and 1 µM rosiglitazone (Abcam, Ab 120762). Induction medium was replaced after two days with maintenance medium containing growth medium supplemented with insulin and rosiglitazone only. Stable 3T3-L1 shRNA lines for knocking down of *Tmem120a* and *Tmem120b* genes were described in[21].

**Human preadipocytes**. Human preadipocytes from formally consenting healthy donors and familial partial lipodystrophy type 2 patients were from the BioLaM biobank (Rizzoli Orthopedic Institute Ethical Committee approval, No. 0018250–2016) located at the CNR Institute of Molecular Genetics Unit of Bologna. Neck adipose tissue of patients undergoing aesthetic surgery had been collected according to local and EU ethical rules. The study design and conduct complied with all relevant regulations regarding the use of human study participants and was conducted in accordance to the criteria set by the Declaration of Helsinki. Pre-adipocyte cultures had been established from the stromal vascular fraction and cultured in D-MEM plus 20% Foetal Bovine Serum and antibiotics. Cells were maintained in D-MEM supplemented with 20% FBS with 100 µg/µl penicillin and 100 µg/µl streptomycin sulphate. Confluent preadipocytes were differentiated into adipocytes in 10% FBS, 0.85 µM insulin, 0.2 nM triiodothyronine (T3), 100 nM pioglitazone, and 125 nM indomethacin as described previously[93].

**Dissections and histology**. Animals were fasting for 16 h, measured and terminated by cervical dislocation. Blood was collected for plasma biochemistry and dissected tissues were weighted and snap-frozen in liquid nitrogen for RNA isolation or fixed in 10% paraformaldehyde (PFA) for 24 h at 4 °C for histology and fluorescent in situ hybridization. PFA-fixed adipose tissue pads were processed in 70% ethanol for 24 h and embedded in paraffin. 4 µm thick sections of adipose tissue were stained by haematoxylin and eosin by the standard protocol for measurement of the lipid droplet size. To measure lipid droplet size in liver, tissues were embedded in OCT compound (*Tissue-Tek*), frozen on dry ice, cut into 4 µm sections and stained with haematoxylin and eosin. Pictures were taken at 40x magnification with the Axio Imager microscope (*Zeiss*) and Qimaging Micropublisher 3.3 camera (*Digital Imaging Systems*). Lipid droplet area was measured by Image J (NIH, USA) software with the Adipocytes Tolls plugin (*Montpellier Ressources Imagerie*). Three animals per group were analysed with 10 random area pictures per animal.

**Metabolic assessment**. 3-4 months old female and male mice were subjected to dietary intervention. In high-calorie diet experiments, animals were fed a 58% kcal from fat, 25.5% kcal from carbohydrates (mainly sucrose) and 16.4% kcal from protein diet (Research Diets, cat. D12331). Low-calorie diet delivered 10.5% kcal from fat, 73% kcal from carbohydrates (mainly corn starch) and 16.4% kcal from proteins (Research Diets, cat. D12328). For pre-experimental maintenance animals were fed standard RM1 diet (SDS Diets, cat. 801151). To assess glucose metabolism, mice underwent glucose tolerance test (GTT). Briefly, 12 h fasting animals were given a bolus intraperitoneal injection of 20% D-glucose solution (SIGMA, G7021) in saline at 2 mg/g body weight. Glucose level was measured in the blood from a tail venesection at time 0, 15, 30, 60, 120 min using an Accu-Check, Aviva Nano, (Roche) glucose monitor. To evaluate whole body insulin sensitivity, an insulin tolerance test (ITT) was performed. After a 6 h fast, animals were injected with a bolus of 0.2 U/mL insulin (Humulin S, Eli Lilly and Company Ltd.) in saline to deliver 1 mU/g body weight. Glucose level during ITT was measured at the same time intervals as in GTT.

**Indirect Calorimetry and TD-NMR**. To evaluate energy expenditure, respiratory exchange ratio ($CO_2/O_2$), water consumption, food intake, activity (X-Y-Z), was measured in indirect calorimetry chambers (PhenoMaster, TSE Systems). Single caged mice were analysed for at least 48 h after at least 24 h of acclimation to their new cage. There was *ad libitum* access to water and food. Oxygen and carbon dioxide levels were measured in real-time by a set of sensors. Fat and lean mass was determined by TD-NMR (Live Mice Analyzer, The Minispec LF90II, Bruker). Energy expenditure is presented as Kcal/kg lean mass/min.

**Plasma biology**. Blood collected to an anticoagulant (EDTA) coated tubes was centrifuged in 4 °C at 2800 rpm for 10 min and supernatant collected and snap-frozen in liquid nitrogen. High Molecular Weight (HMW) Adiponectin ELISA (ALPCO, cat. 47-ADPMS-E01, RRID:AB_2801466) was used according to manufacturer instruction. HMW fraction was used as it is a better indicator of insulin sensitivity for health and disease status monitoring[101]. Insulin and Leptin ELISA were used according to manufacturer's instructions (Chrystal Chem Inc., ref. 90080 - RRID:AB_2783626 and 90030 - RRID:AB_2722664 respectively). To measure the level of triglycerides LabAssay Triglyceride kit was used according to manufacturer instructions (Fujifilm Wako Diagnostics, cat. 290-63701). Non-esterified fatty acids (NEFA) levels were measured by an enzymatic colorimetric assay (Fujifilm Wako Diagnostics, cat. HR Series NEFAHR2, No. 99934691), using manufacturer manual. Plasma biology ELISA experimental data were acquired by Multiplate reader, Multimode Plate Reader—Molecular Devices M5, SoftMax Pro software, Version, 6.1.

**Western blotting**. For Western blotting cell pellets and ground tissue were suspended in SDS 2x sample buffer, sonicated and heated for 5 min at 95 °C. Next proteins were separated by SDS-PAGE and transferred to nitrocellulose membranes. Those were then blocked with 5% BSA in T-BST buffer for 30 min at room temperature and probed with primary antibody overnight at 4 °C. After washing secondary antibodies were loaded for 1 h at room temperature. Finally, membranes

were visualised by Li-Cor Odyssey CLx system. Primary antibodies used were as follows: rabbit anti-Tmem120a at concentration 1:200 (R1196), rabbit anti Tmem120b at concentration 1:200 (R3246), rabbit anti-GFP at concentration 1:500 (Invitrogen, A11122) and mouse anti-γ-Tubulin at concentration 1:1000 (SIGMA, T6657). Secondary antibodies were donkey anti-rabbit IRDye 800CW (Cat. 926-32213) at concentration 1:5000 and donkey anti-mouse IRDye 680RD (Cat. 926-68073) at concentration 1:5000; both from Li-Cor. Representative pictures of uncropped and unprocessed membranes are supplied in Source Data file.

**Stromal vascular fraction (SVF) and adeno-associated virus (AAV)**. To obtain primary, preadipocyte-rich cell population, stromal vascular fraction (SVF) cells were isolated from inguinal subcutaneous fat pads. Three-months-old mice, homozygous for *loxP/loxP* were terminated, fat pads removed and washed in PBS. Tissue was cut into 3–4 mm cubes and placed in digestion medium: KREB's Phosphate Buffer (118 mM NaCl, 5 mM KCl, 1.2 mM $MgSO_4$, 12 mM $Na_2HPO_4$, adjusted to pH 7.4 with 100 mM $NaH_2PO_4$) supplemented with 2 mg/mL collagenase I (Gibco, cat. 17018-029,), 1% Bovine Serum Albumin, 1 mg/mL glucose and 1.265 mM $CaCl_2$. Diced tissue was incubated for 1 h at 37 °C with slow agitation. The reaction was stopped by adding SVF maintenance media (DMEM/F12 media with 10% FBS, 100 U/mL penicillin, 100 µg/mL and 2 mM L-Glutamine). Released cells were immediately centrifuged at 1500 rpm for 5 min to separate SVF cells from mature adipocytes. The SVF cells in the pellet were resuspended in the maintenance media, passed through a 70 µm cell strainer, centrifuged again, resuspended and plated in fresh SVF maintenance media at a concentration of 500 cells/cm$^2$. SVF cells were differentiated into adipocytes under the same protocol as used for 3T3-L1 cells described above. To achieve *Tmem120a* knockout in primary SVF cells *loxP* sites recombination was induced in vitro by the delivery of the *Cre* recombinase gene fused with GFP in AAV virus (RRID:Addgene_49056). As a control AAV with the GFP-only construct was used (RRID:Addgene_49055). Virus particles were produced in HEK293FT (ATCC, Cat. PTA-5077, RRID_6911) cells by co-transfection of AAV plasmids with the helper plasmid pAdDeltaF6 (RRID:Addgene_112867) and AAV packaging plasmid expressing Rep2/Cap5 genes for production of serotype 5—pAAV2/5 (RRID:Addgene_104964).

**Mitochondrial Respiration (Stress Test and Substrate Dependency)**. Oxygen consumption rates (OCR) from differentiated SVF cells were analysed by the Seahorse XF24 Extracellular Flux Analyzer - (Agilent Technologies) with the Wave Controller Software, v:2.4. Cells were seeded, infected with AAV-Cre virus and differentiated into adipocytes on V28 Seahorse plates. Until fully differentiated (day 7-8), cells were cultured at 37 °C with 5% $O_2$ and 5% $CO_2$ atmosphere. 30 min before the experiment, maintenance media was replaced with XF base medium (Agilent Technologies cat. 103193-100) supplemented with 2 mM L-glutamine, 2 mM pyruvate and 10 mM glucose for pre-incubation in a non-$CO_2$ atmosphere at 37 °C to equilibrate the cells with the XF media. V28 plates were used because SVF cells are known to have higher respiration than other commonly cell types (e.g., fibroblasts, cancer cells) and these plates adjust the probe position so that the higher readings do not saturate the machine. Basal and maximal respiration, as well as ATP-linked and proton leak respiration, were determined with a mitochondrial stress test, using optimized drug concentrations. First, oxygen consumption rate (OCR) was measured three times for calculation of basal respiration, followed by an injection of 4 µM oligomycin (SIGMA, 75351) to calculate respiration linked to ATP production. Next, 1 µM FCCP (respiratory uncoupler, SIGMA, C2920) was added for calculating maximal respiration. Finally, simultaneously, 1 µM of rotenone (SIGMA, R8875) and 1 µM of antimycin A (SIGMA, A8674)—blockers of electron transport chain complex I and III, respectively, were added to terminate mitochondrial respiration. That enabled the correction of basal and maximal respiration by subtraction of non-respiratory OCR. ATP linked respiration was calculated by the drop in OCR following oligomycin. Proton leak was calculated by subtracting ATP-linked respiration from basal respiration. Pyruvate dependency for respiration in *Tmem120a* knockout adipocytes was performed in the same assay media as mitochondrial stress assay. Briefly, after basal OCR measurement, 6 µM of mitochondrial pyruvate transporter inhibitor UK9055 (Merck, cat. 5048170001) were added to test if mitochondria have the capacity to utilize other substrates than pyruvate. This was followed by 4 µM oligomycin, to test ATP- production-linked respiration in the absence of pyruvate. Next, 1 µM FCCP was added followed by the simultaneous injection 1 µM of rotenone and 1 µM of antimycin A. Dependency on pyruvate was defined as the difference between the OCR calculated from the vehicle and the UK5099-inhibited OCR for each parameter. Results of the experiments are presented as the oxygen consumption rates (OCR) in pmoles $O_2$ / min / total protein. Protein content in each well was evaluated by the SRB assay (Sulphorhodamine B, SIGMA, cat. S1402) as described elsewhere[102].

**qPCR**. For isolation of total RNA from cultured preadipocytes, adipocytes and mouse adipose tissue RNeasy Lipid Tissue Mini Kit was used (Qiagen, cat. 74804) following manufacturer instruction. Quality of obtained RNA was tested on the capillary electrophoresis machine (2100 Bioanalyzer, Agilent) and material with RIN ≥ 7.0 (RNA integrity number) was used for further applications. 1 µg of RNA was used for cDNA synthesis and qPCR reaction was performed in LightCycler480II instrument with SYBR Green I Master Mix (Roche) and with following

primers: mouse *Tmem120a* forward: 5′ GAAAACCAGATGAAAGAGCGC and *Tmem120a* reverse: 5′ GTGAAGGAGATGACGATGAGG. Expression was normalized to mouse *Gapdh*: forward 5′ CCTCGTCCCGTAGACAAAATG, reverse 5′ TTGACTGTGCCGTTGAATTTG and mouse *Rsp29*: forward 5′ GAGCCGACT CGTTCCTTT, reverse 5′ TGTTCAGCCCGTATTTGC.

To assess the miRNA profile in subcutaneous adipose tissue semi-high throughput, qPCR based, miRNome profiler was used, following manufacturer instructions (QuantiMir, System Biosciences, cat. RA670A-1). To acquire the miRNA-enriched RNA fraction from adipose tissue miRNeasy Mini Kit was used (Qiagen, cat. 217004), accordingly to producer manual in LightCycler480II instrument with SYBR Green I Master Mix. Experiments were performed in 5 biological replicates. Data were analysed in R as per BioConductor HTqPCR package, version:1.48[103]. Briefly, Ct values were normalized per plate to the geometric mean of a set of provided house-keeping genes, and then normalized per sample using the 'quantile' function. Statistics were calculated using Limma[104] and miRNA genes were ranked by their p-values. We selected both genes that ranked in the top 15%, and those that were differentially expressed 2-fold or more and used the combined set for downstream analysis.

**RNA-seq**. Total RNA library was constructed with the TruSeq Stranded Total RNA Gold kit (Illumina, cat. 20020598), which is a ribosomal depletion-based kit. The RNA-seq experiment was executed by the North Texas Genome Centre at the University of Texas (Arlington, US), in the NovaSeq6000 machine, generating sequence dept of approximately 120 M + 120 M (paired) reads. 150 bp paired-end reads were aligned to the mouse mm9 assembly using STAR (v2.5.3.a, RRID:SCR_015899). The resulting alignments were filtered for reads that were properly paired, uniquely mapped and outwit blacklisted regions in the mm9 genome using BEDtools (v2.2.7, RRID:SCR_006646) and SAMtools (v1.9, RRID:SCR_002105). Pseudo alignment of read to transcripts was performed with Salmon (v0.13.1, RRID:SCR_017036) using cDNA and non-coding RNA sequences obtained from Ensembl NCBIM37.67 release. Read counts per gene were subsequently calculated using the R package tximport (v. 1.22) and Ensembl gene annotations (release 67). DESeq2 (v1.24.0) was used to find differentially expressed (DE) genes between pairs of samples, using an adjusted p-value of 0.1. Gene ontology (GO) analysis of the DE genes was performed with the web-based tool g:Profiler (RRID:SCR_006809)—https://biit.cs.ut.ee/gprofiler/gost[105]. Tissue-specific gene enrichment analysis was evaluated by the web tool TissueEnrich—https://tissueenrich.gdcb.iastate.edu[106].

**Microarrays**. 3T3-L1 cells stably transfected with *Tmem120a*, *Tmem120a/Tmem120b*, or control non-target shRNAs were induced to differentiate for 6 days and RNA was extracted with Tri-Reagent (Sigma) according to the manufacturer's instructions. The RNA was converted to cRNA and labelled with biotin using the Illumina TotalPrep RNA Amplification Kit (*Ambion*, AMIL1791). For each analysis, at least two biological replicates were hybridised to Illumina whole-genome gene expression arrays (MouseWG-6 v2.0 BeadChip). Hybridisations were carried out by the Wellcome Trust Clinical Research Facility associated with the University of Edinburgh, using an Illumina Beadstation. Microarray data were quantile normalised and analysed in the R environment using the Bioconductor package Limma (RRID:SCR_010943)[104] using pre-adipocyte 3T3L1 cells as a reference. We selected differentially expressed transcripts using moderated F-statistics and adjusted for a false discovery rate of 5%[107]. A low intensity filter was applied, eliminating transcripts that do not reach a log2(signal) threshold of at least 6.5 in every sample, and we chose a low fold-difference cut-off of 1.4 (abs(log2 ratio)=0.5)

**DamID**. DamID was performed as in[43]. Briefly, 3T3-L1 pre-adipocytes or 6-day differentiated adipocytes ± *Tmem120a* knockdown were transduced with either Dam-LaminB1- or Dam-only encoding lentiviruses and maintained for 3 days in the presence of 10 ug/ul protamine sulphate. At day 9 of differentiation genomic DNA from transduced cells was extracted and processed into libraries for next generation sequencing (Beijing Genomics). Sequenced reads were mapped to mouse mm9 genome and the log2(LaminB1/Dam) value determined for all genomic *DpnI* fragments. LADs and differential regions (IP and PI) were determined as described in Robson et al.[59]. Briefly, LADs were determined using the peak finder function in the BioConductor package DNAcopy (Seshan VE and Olshen A, 2016) *DNAcopy: DNA copy number data analysis*. R package version 1.46.0, (RRID:SCR_001905). IP and PI regions were then identified as regions displaying differential LADs.

**3D fluorescent in situ hybridization (FISH) and gene positioning analysis**. Probes for immuno-FISH on mouse cells and tissues were digoxigenin-labelled (Digoxigenin-11-UTP, *Sigma*) by nick translation from the following BAC or FOSMID clones covering genes loci: *Dctd*—RP23-441F24, *Fh1*—RP23-129O14, *Myh1* and *Myh2*—WI1-2202A13, *Ano5*—RP24-232N14, *Myo18b*—WI1-1944G3, *Unc13c*—RP24-294O13 (all probes from BACPAC Genomics). Human pre-adipocytes and adipocytes were FISH examined with the probes generated from the following BACs: TRIM*55* - RP11-164L7; *MYH1* and *MYH2* - RP11-964J13; *ANO5* - RP11-116B7 and *MYO18B* - CH17-161F16 (all from BACPAC Genomics).

To prepare the probe for hybridisation ~200–600 ng of labelled probe DNA was ethanol precipitated with the 5 µg of sheared salmon sperm DNA and 5 µg of mouse or human *Cot1* DNA (both from *Invitrogen*) and resuspended in hybridisation buffer (50% formamide, 2x SSC, 1% Tween20 and 10% dextran sulphate). After incubation at room temperature for 1 h, the dissolved probe was denatured at 80 °C for 5 min, slowly re-annealed at the 0.1 °C/s speed and kept at 37 °C until the specimen was processed. 4 µm thick paraffin sections of adipose tissue were deparaffined by incubation in 60 °C for 15 min, then immersed in HistoChoice Clearing Agent (Sigma, cat. H2779) twice for 3 min. After that, tissues were rehydrated in decreasing concentrations of ethanol; 100%, 90% and 70%, 3 min each. Next, membranes were permeabilised in 20 µg/mL proteinase K solution in 50 mM Tris for 15 min. at 37 °C and slides rinsed 5x in distilled water. Subsequently, adipose tissue samples were permeabilised in 2 M ice-cold acetic acid for 20 s and washed with water 5x. Similarly, cultured on coverslips adipocytes were fixed in 4% PFA for 10 min and permeabilised in 10 µg/mL proteinase K for 2 min at 37 °C. Preadipocytes were permeabilised in 0.5% Triton-X in PBS for 5 min. After permeabilisation tissue sections, cultured pre- and adipocytes were treated in the same way. First, samples were blocked in 2% BSA and 0.1% Tween20 solution in PBS for 30 min at room temperature. Samples were then incubated with rabbit polyclonal primary antibody against A-type lamins[108] in blocking solution, followed by the donkey anti-rabbit 594Alexa Fluor secondary antibody (Invitrogen, RRID:AB_141637). After washing, immunofluorescence staining was fixed in 2% PFA for 1 min, samples were washed and incubated for 1 h with 100 µg/mL RNase A at 37 °C. Coverslips were then dehydrated in increasing concentrations of ethanol and denatured for 20 min in 70% formamide. After final dehydration labelled probe was loaded on the specimen and hybridised overnight at 37 °C. The unbound probe was then washed, hybridisation signal visualised by the 488Alexa Fluor-conjugated anti-digoxigenin antibodies (Jackson ImmunoResearch, Cat. 200-542-156 RRID:AB_2339039) and nucleic acids were stained by Hoechst 33342 (Thermo Scientific). Finally, coverslips or slides were washed and mounted with Vectashield medium (Vector Laboratories). Image stacks (0.2 µm steps) were obtained using a TE-2000 microscope (*Nikon*) equipped with a 1.45 NA 100x objective, Sedat quad filter set, PIFOC Z-axis focus drive (*Physik Instruments*), and Prime 95B camera (Photometrics) run by MetaMorph, v- 7.8 (Molecular Devices) image acquisition software. Image stacks were deconvolved using AutoQuant X3 (Media Cybernetics, RRID:SCR_002465) software. Micrographs were saved from source programs as 12-bit.tiff files and prepared for figures using Affinity Designer (Serif, RRID:SCR_016952) and Photoshop (Adob*e*, RRID:SCR_014199). Loci signal distance from the NE was measured with the Imaris software (Oxford Instruments, RRID:SCR_007370). A minimum of 50 nuclei were analysed per animal/condition.

**Statistics and reproducibility**. Statistical analysis was performed using GraphPad Prism versions from 8 to 9 and by R software. In microarray experiments differentially expressed transcripts were selected by using moderated F-statistics using Limma and were selected for a false discovery rate (FDR) of 5% and log2(fold change) > 0.5. Also, HTqPCR statistics were calculated using F-statistics by Limma and miRNA genes were ranked by their p-values. Two groups of biological data were analysed by two-sided Student's *t* test. Power calculations and computation of sample size was performed a priori by the G*Power v3.1 software suite with predetermined effect size. Data plotted as graph bars with dots for each data point, represent means with SEM. Box plot graphs were generated by the Tukey method represent median as line and mean as cross. Whiskers in box plots represent data below 25[th] percentile and above 75th percentile and outliers are represented as points outside of the whiskers. For FISH experiment analysis 50 nuclei were analysed per animal and similarly 50 cells were analysed for each technical replicate where cell lines were used. For animal studies two independent experiments were performed. For FISH on cell lines data were collected from two technical replicates. Experiments on subcutaneous primary cells were performed on two animals each divided into two groups for 6 independent adenovirus infections each. Analysis of genome organisation primary cells from human participants was performed once for each gene loci on adipocytes and preadipocytes.

**Bioinformatics analysis**. Gene ontology analysis to test for over-representation using a hypergeometric test was performed using the Bioconductor package GOstats (RRID:SCR_008535)[109]. Only significantly over-represented GO terms (P < 0.002) were considered for further analysis. Biological Process terms associated with tissue remodelling, morphogenesis, development, and cell differentiation were grouped as "development"; terms involving localisation, cell locomotion, chemotaxis, migration and motility were grouped as "cell locomotion/ migration"; and terms including proliferation or growth were grouped as "cell growth/ proliferation". Tissue specific enhancers annotations were sourced from Enhancer Atlas 2.0 www.enhanceratlas.org.

**Materials availability**. Mice generated in this study will be deposited in Charles River laboratories. Plasmids generated for this study will be deposited in Addgene. A-type lamin (3262), Tmem120a and b (R1196 and R3246 respectively) antibodies are not commercially available and will be available upon request. Stable cell lines

will be available upon request. Human samples might be available upon ethical approval. No other unique reagents were generated for this research.

**Reporting summary**. Further information on research design is available in the Nature Research Reporting Summary linked to this article.

## Data availability

The data that support this study are available from the corresponding author upon reasonable request. Raw datasets and bioinformatic analysis of data used in this publication are available through public repositories. RNA-Seq datasets have been made available through the Gene Expression Omnibus (GEO) repository at NCBI under accession number GSE147016, DamID data is available through the Gene Expression Omnibus (GEO) repository at NCBI with accession number GSE150053. 3T3-L1 transcriptome data (microarrays) is available through GEO repository number GSE150022 and intersection of DamID and microarray under the number GSE150507. Supplementary Data files can be found online with the manuscript and at https://data.mendeley.com/datasets/8hgntm9x3w/draft?a=5fef1d20-884f-4237-8686-1ecb1c02e47c (version 2). Source data are provided with this paper.

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

## Acknowledgements

We thank Sue Shackleton (Leicester) for helping us establish the 3T3-L1 system and David Kelly for assistance with image analysis. We also thank Honours student Jeremy Stewart who did the first experiments indicating the project was viable. D.G.B. was supported by a Darwin Trust Studentship and this work was principally supported by Wellcome Senior Research Fellowship 095209, Medical Research Council MR/R018073, and Muscular Dystrophy UK 18GRO-PG24-0248 to E.C.S., Wellcome Trust New Investigator Award 100981/Z/13/Z to N.M.M. and Wellcome Trust Centre for Cell Biology core funding (092076).

## Author contributions

R.C., D.G.B., R.N.C., M.S., A.S., C.R.D. conducted and designed experiments. G.L. provided patient samples. J.I.H., S.W. conducted bioinformatic analysis. E.C.S., R.C. and N.M.M. wrote manuscript.

## Competing interests

The authors declare no competing interests.
