## [Peer Review File · Nature Communications]

REVIEWER COMMENTS

Reviewer #1 (Remarks to the Author):

In this manuscript, Czapiewski and colleagues reported the generation and characterization of the adipocyte-specific *Tmem120a* knock-out mice. Female mutant mice develop metabolic abnormalities under high-fat diet including lipodystrophy, insulin resistance and changes in cellular respiration. The authors claimed that the phenotype is caused by gene expression changes occurring in mutant animals due to the aberrant repositioning of genes, enhancers and miRNA loci between the nuclear interior and the periphery.

The study reveals important and original functions of *Tmem120a*. However, the mechanism of how this protein regulates the changes in chromatin and gene positioning is still not clear. A number of questions should be addressed before publication.

Main comments:

1. The authors showed a wide range of molecular defects in the *Tmem120a* knock-out mice, including transcriptional changes, microRNA deregulation and chromatin alterations. However, it is very surprising that mice under chow diet or even males under high-fat diet are overtly normal. It is known that many metabolic diseases show sexual dimorphism in both human and mouse, but the authors should clarify why it is happening in their model. Are there differences in *Tmem120a* expression between male and female mice? What is the sex of the mouse 3T3-L1 adipocyte cell line authors used?
2. The statement that *Tmem120a* knock-out mice have latent lipodystrophy indicates that *Tmem120a* functions only under certain stress conditions (High-fat diet). This is contradictory with the proposed global role in controlling chromatin organization. The authors should show that *Tmem120a* has roles under other insults. In this regard, a high-fructose diet, which has been proven to cause upregulation of lipogenic pathways and insulin resistance, could be used to challenge the *Tmem120a* knock-out mice which also undergo defects in carbohydrate utilization.
3. The authors show the expression pattern of *Tmem120a* in different mouse tissues. However, the study is missing localization experiments to prove that *Tmem120a* is restricted to the nuclear envelope. Importantly, a recent report (Beaulieu-Laroche et al., 2020) demonstrated that the isoform expressed in the brain is a cell membrane ion channel involved in mechanotransduction. Why is it different in adipose tissue? What is the localization of *Tmem120a* in the fat? Is it solely on the nuclear envelope or also in other cell membranes (plasma membrane, ER, mitochondria, ...)?
4. *Tmem120b* is the mouse paralog of *Tmem120a*. Although the authors performed some experiments using KO cell lines for both genes the degree of functional redundancy between them needs to be further clarified.
5. According to the authors, one reason that accounts for the lipodystrophic phenotype is the changes on microRNA expression. However, this is a correlational observation since several hundreds of mouse microRNAs have been identified, so changes in the expression affecting a small group of them may not be biologically relevant. The authors should at least show that the expression of microRNA-target genes changes accordingly.
6. The authors proposed the *Tmem120a* knock-out as a model for FPLD2 and showed that there are alterations in gene positioning in patient cells. How is the expression on *TMEM120A* in those patients?

Minor comments:

1. The authors should indicate the Tmem120a and Tmem120b protein expression levels (measured by WB) in their conditional KO mice.
2. How do the authors explain that despite brown adipose tissue of mutant mice accumulating much more fat than WT, there is no difference in BAT deposits weight?
3. The FISH experiments clearly show the aberrant localization of some genes in mutant mice. However, there is variability regarding the number of copies found among the different experiments. Theoretically, and assuming euploidy, there should be either two or four (if the gDNA has been duplicated) copies per cell. Why in some panels there is one or three copies? Is it because they co-localize?

Reviewer #2 (Remarks to the Author):

This manuscript is a tour-de-force study of 3D genome control by Tmem120a, an inner nuclear membrane protein highly expressed in fat cells. Adipocyte-specific loss of Tmem120a in mice alters the nuclear envelope association (3D genome positioning) of enhancers and genes encoding proteins and specific lncRNAs that specifically control fat cells and muscle cells. These careful, rigorous experiments suggest Tmem120a functions as a 3D-genome 'switch' that couples the promotion of adipocyte functions with the suppression of muscle functions. Tmem120a-dependent chromatin control is physiologically relevant as shown by detailed analysis of metabolic phenotypes in mice, and by comparison to cells from patients with Familial Partial Lipodystrophy (FPLD2), the best-characterized metabolic disease caused by dominant missense mutations in A-type lamins (LMNA). Overall this is a rigorous and convincing genomic, cellular and physiological analysis of Tmem120a. It provides innovative Tmem120a-based mechanistic insight into FPLD2 phenotypes, which include both the gain and loss of different fat 'depots' (white / beige / brown) as well as significant gain of muscle tissue. These findings, especially the discovery that Tmem120a regulates the expression of another major group of 'upstream' regulators (lncRNAs that regulate fat-specific versus muscle-specific genes), provide a major advance in understanding how this (and possibly other) nuclear envelope transmembrane ('NET') proteins influence 3D-genome architecture as upstream control points for complex tissue-specific diseases.

The manuscript requires correction of one figure, and essential clarifications to text, detailed below.

Figure 7e: For this figure to accurately depict Tmem120a at the inner membrane (only) of the nuclear envelope, change "cytoplasm" (right panel) to "lumen". Without this change, the middle panel shows individual Tmem120a proteins each spanning the entire nuclear envelope (inner membrane, lumen, outer membrane) and contacting both chromatin and cytoplasm.

Text clarifications:

- (1) Confusing and inaccurate terminology for A-type lamins ("laminA") in the abstract ("...FPLD2 patients that harbor nuclear envelope protein laminA mutations") and throughout the manuscript.
 - (a) Calling laminA a "nuclear envelope protein" misleadingly fails to distinguish lamins (nuclear intermediate filament proteins) from Tmem120a and other true integral membrane proteins.
 - (b) The term "laminA" should be replaced systematically with either "A-type lamins" (when referring

to proteins) or LMNA (when referring to the gene or disease-linked mutations). This is essential for accuracy because LMNA encodes a second abundant protein, lamin C, which forms independent filaments with overlapping and unique roles, and is also affected by most FPLD2-causing mutations in LMNA.

(c) Need to introduce "LMNA" gene and "A-type lamins" in paragraph 2 of Introduction.

(b) Last sentence of abstract confusingly states that the normal function of Tmem120a is to "instigate dysfunction". Re-phrase in positive terms.

(c) First paragraph of Introduction: First and last sentences are confusing and vague.

(d) Page 5, paragraph 2: For the discussion of Tmem120a [mRNA or protein?] expression in Arizona Pima Native Americans and other humans:

(a) Present these results first (not mixed among results from mouse models);

(b) Change "was downregulated", a verb often used to describe an experimental manipulation, to a simple summary of the findings, e.g., "was ___% lower than control populations".

(c) Add the genetic/ethnic identifiers for the population(s) referred to as "in obesity with diabetes (data obtained from the GEO dataset)"

(d) Change "stronger effects in females" to "stronger effects in women" [or women and girls, if girls were included in the GEO dataset].

Page 5, simplify the last sentence of paragraph 2. E.g., "accumulation, with hypothesized roles in nociception in brain." (?)

Page 5, edit and simplify the last two sentences for clarity.

Page 7, bottom paragraph, line 2: change "on HFD was accounted by reduced cell numbers (hyperplasia) or reduced fat accumulation (hypertrophy)" to "on HFD was due to fewer cells (should this be "hypoplasia", rather than "hyperplasia"?) or reduced fat accumulation per cell ("hypertrophy" seems incorrect here)"

Page 8, lines 2-3: I would not describe a 1-cm change in body length as "small"; indicate the percentage reduction. Do the reduced body and tail lengths warrant comment or speculation as a potential impact on skeletal development?

Page 9, second paragraph, first sentence is unclear ("constrained adipose expansion"?).

Page 9, sentence that begins "Hyperlipidemia has been reported.." is overly long; needs editing for clarity.

Page 9, last sentence, define "polydipsia" and change "worsened" to "more severe".

Page 10, confusing first sentence ("To determine.."). E.g., to determine if Ad-Tmem120a^{-/-} female mice accumulated less fat because they....., we performed...?"

Page 15, middle: where is the data supporting this statement? "Tmem120a knockdown was 90% compared to..".

Supplementary Figures: If there are only 3 supplemental figures, why are they numbered 1, 3 and 5 instead of Supplemental Figures 1, 2 and 3?

Response to reviewer's comments

We thank both reviewers and editors for their time, comments and suggestions that have improved our manuscript. In this response to reviewers, we address every comment from Reviewers #1 and #2 and describe the new data we have added and changes to figures and the text made to clarify or improve the manuscript where needed. We believe that the additional data shown and text changes both solidify our results and should clarify the manuscript on points where we were not sufficiently clear in the original version and hope that you agree and will now accept the manuscript as suitable for publication in *Nature Communications*. Below the reviewer comments are copied in blue italics followed by our responses in black text and specific textual changes in dark grey italics.

Reviewer #1 (Remarks to the Author):

In this manuscript, Czapiewski and colleagues reported the generation and characterization of the adipocyte-specific Tmem120a knock-out mice. Female mutant mice develop metabolic abnormalities under high-fat diet including lipodystrophy, insulin resistance and changes in cellular respiration. The authors claimed that the phenotype is caused by gene expression changes occurring in mutant animals due to the aberrant repositioning of genes, enhancers and miRNA loci between the nuclear interior and the periphery.

The study reveals important and original functions of Tmem120a. However, the mechanism of how this protein regulates the changes in chromatin and gene positioning is still not clear. A number of questions should be addressed before publication.

We thank the reviewer for this positive comment. We believe we are able address all the questions with additional data we have added and/or clarification.

Main comments:

1. The authors showed a wide range of molecular defects in the Tmem120a knock-out mice, including transcriptional changes, microRNA deregulation and chromatin alterations. However, it is very surprising that mice under chow diet or even males under high-fat diet are overtly normal. It is known that many metabolic diseases show sexual dimorphism in both human and mouse, but the authors should clarify why it is happening in their model. Are there differences in Tmem120a expression between male and female mice? What is the sex of the mouse 3T3-L1 adipocyte cell line authors used?

First and foremost, in answer to the question about Tmem120a expression differences in male and female mice, indeed there is roughly a 4-fold higher expression of Tmem120a in females over males. We thank the reviewer for asking this question as it provides a simple explanation for much of the sex differences. Accordingly, we have now added this data to **Supplementary Figure 1 panels i and j**.

It is thus not surprising that males do not show any statistical differences in body weight between the wild-type and knockout whereas females had a strong effect. Although there is no body weight effect in males, we tested glucose tolerance and insulin tolerance in males and, though we have no significant difference in glucose tolerance on either high- or low-fat diet, we find the males to be more insulin resistant on high-fat diet. Thus, the loss of the minimal Tmem120a expressed in males does indeed have some consequences for metabolism. This data has now been added in the **new Supplementary Figure 3** and we have described this in the text as *“Interestingly, regarding sex differences, male Ad-Tmem120a^{-/-}*

mice showed only moderate differences that were not statistically significant in GTT, but had similarly strong effects as females on ITT (Supplementary Fig. 3)."

Regarding the sex of 3T3-L1 cells, this is a well-established adipogenesis model that we got from the ATCC that was clonally derived from 3T3 cells that originated from fibroblasts of Swiss Albino mouse strain embryo and historically thought to be female. However, there is a somewhat confusing literature addressing this question in more recent years. In one study it was clearly shown that there are two X chromosomes, but Y chromosomes were identified that were thought to be "derivative" containing characteristics of chromosome 7 in the progenitor 3T3 cells (Leibiger et al., 2012). A more recent publication determined this progenitor 3T3 cell line as of female origin by qPCR (Wan et al., 2014). Thus, it appears that they are female and so should well recapitulate the stronger effects of the female mice in the *in vitro* system.

2. The statement that Tmem120a knock-out mice have latent lipodystrophy indicates that Tmem120a functions only under certain stress conditions (High-fat diet). This is contradictory with the proposed global role in controlling chromatin organization. The authors should show that Tmem120a has roles under other insults. In this regard, a high-fructose diet, which has been proven to cause upregulation of lipogenic pathways and insulin resistance, could be used to challenge the Tmem120a knock-out mice which also undergo defects in carbohydrate utilization.

The reviewer has picked up on one of the points that makes our study very interesting and important and one that we fear we did not describe sufficiently clearly to get this comment. Tmem120a does not function only under the high-fat diet condition or only in female mice, but it directs genome organisation in differentiation in all fat cells regardless of sex or diet. Yet the lipodystrophy pathology is only penetrant in females and on the high-fat diet. We have now included data confirming that the genome organisation changes on the low-fat diet. While this may seem counterintuitive at first, this is the data and so the question is why is the phenotype only penetrant under high-fat diet? We think that this is another way in which the mouse mirrors the human nuclear envelope-linked lipodystrophy. In the human disorder there is typically no indication of disease at an early age and the pathology begins to present in girls at puberty, when in addition to the big changes in hormones there is also an increased demand on metabolic pathways. Interestingly, the increased deposition of pre-pubertal fat and its associated increase in leptin release, the adipocyte hormone which gates the onset of puberty (Ahima et al., 1997), is an additional burden on cells with any underlying mispositioning defect (e.g., TMEM120A in our case). The high-fat diet does this while the low-fat diet is the principal treatment used for human lipodystrophy patients and thus by reducing the overall metabolic load the moderate defects in metabolic enzyme expression levels do not result in overtaxing of the system (Brown et al., 2016). The current pharmacological treatment for lipodystrophy is the leptin replacement therapy that in different forms has a similar goal of inhibiting hunger and reducing fuel intake by the patients (Meehan et al., 2015).

We are glad that the reviewer raised this point because it has stimulated us to add this data and to highlight this further in the discussion. Specifically, to test if adipose tissue from KO animals on low fat diet also shows changes in genome organisation, we have performed additional FISH experiments localising two genes and, in both cases, the effect was comparable with that observed on high fat diet. This indicates that genes reposition independently to the diet and purely based on the absence of Tmem120A, which is consistent with our original hypothesis presented in the manuscript that the phenotypic changes are related to the underlying gene misregulation making the animals not capable of dealing with the metabolic load of the high-fat diet in the knockout whereas it is sub-phenotypic penetrance when the metabolic load is low on the low-fat diet. This new data is now added to the manuscript as a text and graphs are included in **Supplementary Figure 5 panel g**. We have added to the text to describe this: "*Importantly, these repositioning changes*

in the mice occurred not only on the HFD but also on the LFD (Supplementary Fig. 5g), indicating that the gene repositioning occurs as a result of the knockout and not as a result of the change in diet. Thus, the phenotypic change must reflect a defect in the ability of the cells to respond to the challenge of the metabolic load of the HFD caused by the many metabolism and other genes altered in the knockout.” We have also added to the discussion “...and the inability of the cells with these multiple alterations in adipocyte gene expression to handle the increased metabolic loads on the HFD.”

Importantly, although the phenotype is not as strong as in high fat diet, we also see some mild lipodystrophic characteristics in mice on low fat diet. For example, we see significantly delayed glucose clearance in low fat diet KO animals when comparing to wild type. This was in the original Figure 2a, but was difficult to see because the low-fat diet data was mixed in with the much stronger high-fat diet. We have now separated this into two separate graphs in the **new version of Figure 2 in panels a and b**, thus making the difference more visible. In the manuscript we called these changes “mild” because they are mild in contrast to the striking differences observed on the high fat diet, however the difference even on low fat diet was statistically significant. Thus, we have also changed the text to note this: *“Female Ad-Tmem120a^{-/-} mice showed a pronounced impairment of glucose tolerance on HFD compared to control littermates that could still be observed, albeit more mildly, on LFD (Fig. 2a-c)”*.

Re the issue of testing other diets, indeed it would have been interesting to do this, but there are several very good reasons why we have not done so: 1) Our high-fat diet is not just high fat, but also high sugar and to be specific - sucrose. We assume that the Reviewer's comment in part reflects the very interesting question of whether the effects of fat and of carbohydrates will be the same for the latent lipodystrophy; however, since they already have high sugar in the high-fat diet, to test this would require not just one but several additional mouse experiments. 2) It directly follows from the first point that the requirements to minimise animal usage – specifically the 3Rs movement in UK biological sciences; Reduce Refine Replace (www.nc3r.org.uk) – counter the reviewer request. 3) Since we already had a high carbohydrate content in our diet, the main advantage that these additional animal experiments would provide would be to understand more about which of the Tmem120A-regulated metabolic genes changing are more relevant to different dietary sugars and fats. While this is clearly a very interesting question and one that shows the value of the mouse we are describing as a model of studying lipodystrophy and general metabolic disorders, it is clearly a separate study that goes well beyond the scope of the study we have presented here.

Regarding the specific reviewer suggestion of the high-fructose diet experiment, it is interesting to note that we would expect to see defects also if we only fed the mice carbohydrates and not fat. It can be seen in our data on respiratory exchange ratio that Tmem120a KO animals preferentially utilise carbohydrates under high fat diet nutritional overload while wild-type mice switch to fat burning on the experimental diet (Figure 3e). Furthermore, this effect on the fuel preference in the whole organism is also shown at the cell energetics level (mitochondrial phenotyping by Seahorse) in primary cells, the first description of this effect in relation to lipodystrophy. Thus, our data very importantly give a strong indication that fuel flexibility is affected in lipodystrophy. We agree that systematic assessment of fuel utilization is of importance for progress in the knowledge on lipodystrophy management but, again, such broad further multi-diet studies are not within the scope of this manuscript.

Finally, we added to the materials and methods section additional information about the composition of the diets used i.e., that carbohydrates in high-calorie diet mostly consist of sucrose and in low-calorie diet carbohydrates are mostly made of corn starch.

3. The authors show the expression pattern of Tmem120a in different mouse tissues. However, the study is missing localization experiments to prove that Tmem120a is restricted to the nuclear envelope. Importantly, a recent report (Beaulieu-Laroche et al., 2020)

demonstrated that the isoform expressed in the brain is a cell membrane ion channel involved in mechanotransduction. Why is it different in adipose tissue? What is the localization of Tmem120a in the fat? Is it solely on the nuclear envelope or also in other cell membranes (plasma membrane, ER, mitochondria, ...)?

Indeed, the reviewer is correct that Tmem120a has multiple subcellular localisations reported. We focused our introduction on the expression and function as opposed to the subcellular localisation because we had already published before that Tmem120a is in the ER as well as in the nuclear envelope, though principally in the nuclear envelope (**Reviewer Only Figure 1**). However, the reviewer is correct that this point should be noted and accordingly, we have added a subclause where Tmem120a is first introduced in the Introduction stating, “*which notably we found in both the nuclear envelope and ER*”.

Reviewers Only Fig 1. Examples of published data on the subcellular localisation of Tmem120a. a, Batrakou et al., 2015. b, Zuleger et al., 2013. c-e, Malik et al., 2010.

The reviewer is also correct in that, while we had previously published the localisation in the 3T3-L1 adipocyte system (Batrakou et al., 2015), we never tried to stain with the antibodies in the sections of subcutaneous fat from the mice. We have now done this and found that we could observe a nuclear rim staining in wild-type mice together with what appears to be ER and even possibly some plasma membrane staining. However, notably, when we stained these antibodies on the Tmem120a knockout fat sections, we could no longer see any nuclear rim staining, but still could see what appeared to be ER and plasma membrane staining, suggesting that at least some of that staining was non-specific (**Reviewer Only Figure 2, panel A**). Notably, these antibodies were very specific by Western blot (**new Supplementary Figure 1 panel f and i and Reviewer Only Figure 2, panel B**), though this is for denatured protein and it is possible that it is non-specifically recognising folded proteins in tissues. Because it is unclear whether the ER and plasma membrane staining observed was due to non-specific antibody interactions, we thought it best to not add this data to the manuscript.

Reviewers Only Fig 2. a, Tmem120a staining of iWAT from WT and KO animals. **b**, specificity of Tmem120 a and b antibodies tested on HEK cells transfected with plasmid carrying Tmem120a and b paralogs labelled by GFP.

With respect to the Beaulieu-Laroche et al., 2020 TACAN paper, we would note that there is reason to suspect their conclusions as this paper was recently challenged by three publications just out in eLife, which reported that they cannot reproduce data presented in the Cell paper. One of these studies was led by the Nobel prize laureate Roderick MacKinnon who stated that Tmem120a “produces heterogeneous conduction levels that are not mechanosensitive and are most consistent with disruptions of the lipid bilayer” (Niu et al., 2021). The other studies were similarly sceptical about the Beaulieu-Laroche claim of Tmem120a as mechanosensing channel (Xue et al., 2021 and Rong et al., 2021). Notably, we showed that there is a different splice variant in brain (Batrakou et al., 2015; and **Reviewer Only Figure 3**; so other functions are not excluded and not conflicting with our results.

Reviewers Only Fig 3. a, Different size of protein is detected in brain and fat by Western blotting. Data published in Batrakou et al., 2015. **b**, expression of Tmem120a protein in different regions of mouse brain show different localisation of the protein in different cell types. Not published data.

4. *Tmem120b is the mouse paralog of Tmem120a. Although the authors performed some experiments using KO cell lines for both genes the degree of functional redundancy between them needs to be further clarified.*

We thank the reviewer for this suggestion which highlights a rather embarrassing oversight: in the originally submitted manuscript we introduced this section noting that comparing the Tmem120a knockdown with the Tmem120a/Tmem120b double knockdown "*allowed us to test for functional redundancy between Tmem120a and its paralog Tmem120b*", yet we only showed a handful of DamID traces comparing them in panels d and e of Figure 5. We have now remedied this by performing an analysis of the entire dataset, finding that roughly a third of the combined targets affected by Tmem120a and/or Tmem120b are just affected by Tmem120a while being unaffected by Tmem120b. Roughly 50% were Tmem120b specific targets and roughly 20% were affected differently by both proteins. This new data has been **added to Figure 5 as panel g** and the text has been modified adding: "*Notably, it could be observed in some of the DamID traces (Fig. 5d,e) that there were differences between the Tmem120a knockdown and the Tmem120a/Tmem120b double knockdown. For Parm1 and Pla2g12a the traces were very similar between the two, suggesting that the change was specific to loss of the Tmem120a protein and Tmem120b had no effect. In contrast, for Cpe and Fh1 the change in the LAD intensified in the double knockdown compared to the Tmem120a knockdown alone. For Dctd and Ak4 there were additional LADs added in the double knockdown that did not occur in the Tmem120a knockdown alone. Finally, for Sema3c the Tmem120a knockdown had no effect at the locus, but the LAD was uniquely contributed by the Tmem120b because the change only occurred at all in the double knockdown. This led us to question the extent to which the targets of Tmem120a and Tmem120b might differ; so we set parameters to search for LADs that appeared equally in the Tmem120a knockdown and the double knockdown sample versus LADs that were unique to the Tmem120a knockdown sample or to the double knockdown sample. This revealed that roughly 1/3 (633) of the affected loci were uniquely affected only by Tmem120a with no further change at the locus when both proteins were knocked down while those only affected by Tmem120b (unique to the double knockdown sample) were close to 50% (886) (Fig. 5g). The smallest group were loci for which Tmem120a and Tmem120b seemingly had different target loci within the same gene accounted for ~20% (388) of the affected loci (Fig. 5g). Thus, there is strong evidence for considerable functional diversity between Tmem120a and Tmem120b targets.*"

While this degree of target segregation opens the way for many future studies investigating the functional diversity between these paralogs, to do more at this juncture would go well beyond the remit of this study. The reality of scientific funding is that we need to publish the data on the Tmem120a knockout animals to get further funding to make another mouse knockout of Tmem120b and we may not be able to get permission to knock out both together because we anticipate that this will have a too severe phenotype in mice. Thus, just as it took dozens of papers investigating functional redundancy between the other nuclear envelope transmembrane SUN1 and SUN2 proteins to get to a point where we understand this only partially, it will likely require a stepwise process to understand functional redundancy between Tmem120a and Tmem120b. We nonetheless agree wholeheartedly with the reviewer that this is a very important and exciting question to address in the future.

5. *According to the authors, one reason that accounts for the lipodystrophic phenotype is the changes on microRNA expression. However, this is a correlational observation since several hundreds of mouse microRNAs have been identified, so changes in the expression affecting a small group of them may not be biologically relevant. The authors should at least show that the expression of microRNA-target genes changes accordingly.*

Thank you for this suggestion. The reviewer is correct that this was a general statement indicating a general correlation only in the original manuscript. We have now intersected the RNA-Seq data with data on the known targets of the miRNAs that were changing in the subcutaneous fat of the knockout mouse. This data, while still just correlative, clearly shows that the targets of the changing miRNAs are also accordingly changing in the fat. We have

added into Figure 6 a new panel f in which for the specific miRNAs for which the traces were shown in panel e we show the targets and the directionality of their changes. Moreover, we have added a **new sheet 3 to Supplementary Table 5** in which we list all the data for the intersect of all 160 of the changing miRNAs. The text has been modified to reflect this as *“In all these cases, the relevant target genes of the miRNAs exhibited corresponding changes in our knockout mouse RNA-Seq data (Fig. 6f). This was not only the case for the specific miRNAs shown, but for the wider set found to change in the knockout (Supplementary Table 5, sheet 3).”*

6. The authors proposed the Tmem120a knock-out as a model for FPLD2 and showed that there are alterations in gene positioning in patient cells. How is the expression on TMEM120A in those patients?

This was a good suggestion and, as we still had lysates from the patient primary cells, we performed the Western blot now shown in **new Supplementary Figure 6**. However, there was no reduction in Tmem120a in the patients compared to the control. This is not contradicting our suggestion that Tmem120a might mediate some of the pathology in lamin-linked lipodystrophy because the mutation is in LMNA and not in Tmem120a. Thus, the hypothesis is that the lamin point mutation alters its interactions with Tmem120a and thus affects genome organisation. We would like to further investigate this hypothesis and have been trying to purify the nucleoplasmic fragment of TMEM120A, but so far we have not been able to get a monotypic species which would be needed for proper kinetic binding studies. Therefore, we would argue that this is beyond the scope of the current study. We also added to the text to explain this *“Notably, these changes in the LMNA lipodystrophy patients paralleling the repositioning defects in our Tmem120a knockout mice are not due to loss of Tmem120a in the patients as levels were unchanged (Supplementary Fig. 6); however, due to extensive interactions of nuclear membrane proteins with the underlying lamina, the mutations in lamin A could still be expected to affect the gene repositioning function of Tmem120a.”* We also elaborated in the discussion adding *“...and the lamina polymer they form under the inner nuclear membrane interacts with many nuclear membrane proteins: indeed we previously demonstrated a probable interaction between Tmem120a and the lamin polymer²⁰.”*

Minor comments:

1. The authors should indicate the Tmem120a and Tmem120b protein expression levels (measured by WB) in their conditional KO mice.

Thank you for this suggestion. We have done this and **added it to Supplemental Figure 1**. Specifically, we have shown RNA-Seq tracks confirming the knockout in panel d, have performed Western blots with TMEM120A antibodies to confirm the knockout in panel h that are quantified in panel i, and now that we have antibodies to TMEM120B we have also performed Western blots on the wild-type and knockout iWAT showing that the level of Tmem120b is not changing in the knockout animals shown in the new panel j. Finally, we added Western blotting description to Methods section.

2. How do the authors explain that despite brown adipose tissue of mutant mice accumulating much more fat than WT, there is no difference in BAT deposits weight?

We think that being of the BAT is not only enabling the remaining cells in these depots to accumulate fat, but that it is depleting the total number of cells in the BAT depots. This is consistent with the reported effects of a number of the changing miRNAs and was noted in the text with the sentence *“Of particular note, knockdown of miR-27b, miR-92a, miR-155, and miR-*

150 have all been shown to enhance browning/ beiging^{63,64,65,66}, one of the Ad-Tmem120a^{-/-} pathologies.”

3. The FISH experiments clearly show the aberrant localization of some genes in mutant mice. However, there is variability regarding the number of copies found among the different experiments. Theoretically, and assuming euploidy, there should be either two or four (if the gDNA has been duplicated) copies per cell. Why in some panels there is one or three copies? Is it because they co-localize?

Indeed, the reviewer is right about the ploidy in the 3T3-L1 system, which is primarily close to tetraploid, but can vary with passage. Thus, it is not always clear with these cells whether not all 4 loci are observed because there has been euploidy or because only a single z-plane is shown and the other loci are out of that focal plane. In the case of the primary mouse tissue and primary human cells, these should be diploid and in this case we never observe more than 2 loci and when we only observe 1 it should be solely due to the other being out of the focal plane of the z-plane shown. Finally, it is true that some loci might colocalise and in this case one would expect twice the intensity at the two colocalising compared to other loci. The reviewer is likely correct that this happens in some cases because when we look at the cell in Figure 5 panel f third from the bottom, we see that the middle spot appears to be larger than the other two observed and so could be two loci colocalising.

Reviewer #2 (Remarks to the Author):

This manuscript is a tour-de-force study of 3D genome control by Tmem120a, an inner nuclear membrane protein highly expressed in fat cells. Adipocyte-specific loss of Tmem120a in mice alters the nuclear envelope association (3D genome positioning) of enhancers and genes encoding proteins and specific lncRNAs that specifically control fat cells and muscle cells. These careful, rigorous experiments suggest Tmem120a functions as a 3D-genome ‘switch’ that couples the promotion of adipocyte functions with the suppression of muscle functions. Tmem120a-dependent chromatin control is physiologically relevant as shown by detailed analysis of metabolic phenotypes in mice, and by comparison to cells from patients with Familial Partial Lipodystrophy (FPLD2), the best-characterized metabolic disease caused by dominant missense mutations in A-type lamins (LMNA). Overall this is a rigorous and convincing genomic, cellular and physiological analysis of Tmem120a. It provides innovative Tmem120a-based mechanistic insight into FPLD2 phenotypes, which include both the gain and loss of different fat ‘depots’ (white / beige / brown) as well as significant gain of muscle tissue. These findings, especially the discovery that Tmem120a regulates the expression of another major group of ‘upstream’ regulators (lncRNAs that regulate fat-specific versus muscle-specific genes), provide a major advance in understanding how this (and possibly other) nuclear envelope transmembrane (‘NET’) proteins influence 3D-genome architecture as upstream control points for complex tissue-specific diseases.

We thank the reviewer for this very positive assessment of our work.

The manuscript requires correction of one figure, and essential clarifications to text, detailed below.

Figure 7e: For this figure to accurately depict Tmem120a at the inner membrane (only) of the nuclear envelope, change “cytoplasm” (right panel) to “lumen”. Without this change, the middle panel shows individual Tmem120a proteins each spanning the entire nuclear

envelope (inner membrane, lumen, outer membrane) and contacting both chromatin and cytoplasm.

Thank you for noticing this: indeed, we made a mistake in labelling of this cartoon. The use of dots in representing the membrane was supposed to represent the polar head groups of a lipid bilayer, yet the last panel incorrectly labelled the outer region as cytoplasm when it should have been lumen. This is corrected now and to make this more clear we have further added to the legend these dots as "lipid bilayer".

Text clarifications:

(1) Confusing and inaccurate terminology for A-type lamins ("laminA") in the abstract ("...FPLD2 patients that harbor nuclear envelope protein laminA mutations") and throughout the manuscript.

Thank you for pointing this out. Indeed, while I have seen it used without the space in some reviews recently, the historical presentation was with a space before the A and so we have now changed this to "lamin A" throughout the text where relevant, though in most cases as per the subsequent comment we have used "LMNA" or "A-type lamins" where appropriate instead.

(a) Calling laminA a "nuclear envelope protein" misleadingly fails to distinguish lamins (nuclear intermediate filament proteins) from Tmem120a and other true integral membrane proteins.

Actually, the operational definition of the nuclear envelope includes the underlying lamin polymer. This follows from the fact that its original isolation by the lab of Gunter Blobel required the structural support of the lamina to keep the fraction intact. However, the point made by the reviewer that nuclear envelope transmembrane proteins are a special class of nuclear envelope protein is reasonable and accordingly we have modified the introduction of Tmem120a to describe it as a nuclear envelope transmembrane protein instead of just a nuclear envelope protein while accurately referring to lamin A as a nuclear envelope protein.

(b) The term "laminA" should be replaced systematically with either "A-type lamins" (when referring to proteins) or LMNA (when referring to the gene or disease-linked mutations). This is essential for accuracy because LMNA encodes a second abundant protein, lamin C, which forms independent filaments with overlapping and unique roles, and is also affected by most FPLD2-causing mutations in LMNA.

Thank you, this is an important point. We had originally tried to keep it simple by using the same simple nomenclature throughout because we assume this study will be of interest to, not only the nuclear envelope and genome organisation fields who are more familiar with the lamins, but also the metabolism field where there might be less background understanding of the nuclear envelope and lamins. However, we can see in retrospect that this actually made it more confusing. Accordingly, we have now carefully checked each reference to lamins for accuracy, replacing with "LMNA" or "A-type lamins" as appropriate.

(c) Need to introduce "LMNA" gene and "A-type lamins" in paragraph 2 of Introduction.

The relevant sentence in the 2nd paragraph has now been changed from:

"Importantly, radial genome organisation is disrupted in diseases linked to NE proteins such as laminA, mutations in which cause familial partial lipodystrophy of the Dunnigan-type (FPLD2)^{7,8}, a more severe lipodystrophy⁹, several metabolic syndromes^{10,11}, and type 2 diabetes^{12,13}."

to:

"Importantly, radial genome organisation is disrupted in diseases linked to NE proteins such as A-type lamins. Mutations in the LMNA gene that encodes A-type lamins (principally lamins A and C) cause familial partial lipodystrophy of the Dunnigan-type (FPLD2)^{7,8}, a more severe lipodystrophy⁹, several metabolic syndromes^{10,11}, and type 2 diabetes^{12,13}."

(b) Last sentence of abstract confusingly states that the normal function of Tmem120a is to "instigate dysfunction". Re-phrase in positive terms.

Thank you for pointing this out: indeed it is TMEM120A loss that would result in adipose dysfunction, not TMEM120A itself. What we were trying to convey was, in addition to its loss yielding defects, that point mutations may yet be discovered that yield adiposity disorders and moreover that point mutations in lamin A leading to adipose dysfunction could have these effects mediated through a disruption of the normal function of TMEM120A since the genome organisation defects phenocopied. This is similar to what we found in several different papers and unpublished data for a muscle-specific NET, NET39, where we first showed its knockdown disrupted muscle-specific genome organization (Robson et al., 2016 Mol Cell) and later found point mutations in unlinked Emery-Dreifuss muscular dystrophy patients that similarly alter genome organisation (Meinke et al., 2020 EBioMed). We further found in unpublished work that wild-type lamin A binds NET39 but muscular dystrophy point mutations specifically yield a loss of NET39 binding. These are obviously too complex arguments to make in the short space of an abstract as well as speculating too much on the further functions of TMEM120A; so accordingly, we have simplified the argument and writing. The last sentence has been changed from:

"Our data suggest TMEM120A may mediate/instigate novel categories of adipose tissue dysfunction across the adiposity spectrum and provide a new miRNA-based mechanism possibly driving the unexplained muscle hypertrophy in human lipodystrophy."

to:

"Our data indicates TMEM120A genome organisation functions affect many adipose functions and its loss may yield adiposity spectrum disorders, including a new miRNA-based mechanism that could explain muscle hypertrophy in human lipodystrophy."

(c) First paragraph of Introduction: First and last sentences are confusing and vague.

This again reflects the dichotomy of audiences for whom this study should be of interest as the first sentence was written by our metabolism collaborator for that audience and the last sentence by myself for the genome audience. However, now that it has been noted I can see how each might not be clear to scientists from other disciplines. Accordingly, the first sentence has been changed from:

"The extremes of adiposity, severe obesity and lipodystrophy, lead to insulin resistance and metabolic disease."

to:

"Human diseases linked to defects in adipose tissue and metabolism cover a wide range, the extremes of which being severe obesity and lipodystrophy. Nonetheless, both these extremes of adiposity lead to insulin resistance and metabolic disease."

and the last sentence from:

"Previous studies have yet to identify proteins controlling adipocyte-specific genome organisation, though such organisation is a critical hierarchical determinant of tissue differentiation and gene expression through directing genes to active/inactive regions and enabling enhancer-gene contacts."

to:

"Despite these clear links to genome organisation, the proteins controlling adipocyte-specific genome organisation have yet to be identified. Doing so could yield significant benefits to understanding and treating adipose disease because genome organisation contributes to tissue differentiation and gene expression through directing genes to active/inactive regions within the nucleus and enabling enhancer-gene contacts."

(d) Page 5, paragraph 2: For the discussion of Tmem120a [mRNA or protein?] expression in Arizona Pima Native Americans and other humans:

These data refer to mRNA levels and this information is now added to the sentence.

(a) Present these results first (not mixed among results from mouse models);

Thank you for this suggestion, we have changed the order as suggested with the human results now presented before the mouse results and it definitely improves clarity.

(b) Change "was downregulated", a verb often used to describe an experimental manipulation, to a simple summary of the findings, e.g., "was ___% lower than control populations".

This sentence is now changed as suggested with the reduction in expression given as %.

(c) Add the genetic/ethnic identifiers for the population(s) referred to as "in obesity with diabetes (data obtained from the GEO dataset)"

The ethnic population was Asian Indian and this has been added.

(d) Change "stronger effects in females" to "stronger effects in women" [or women and girls, if girls were included in the GEO dataset].

This change has now been made. The parts of the paragraph encompassing this and the previous 4 corrections reads now as follows:

"In humans, the mRNA level of TMEM120A was approximately 25% lower in adipose tissue of obese Arizona Pima Native Americans²¹ and 40% lower in obesity with diabetes in Asian Indians (data obtained from GEO dataset, GDS3665). Both studies showed stronger effects in women. In the mouse 3T3-L1 adipogenesis tissue culture system, Tmem120a gene knockdown in preadipocytes reduced adipogenesis and lipid accumulation ~30% and combined knockdown with its paralog Tmem120b reduced it ~60%²⁰. Notably, TMEM120A re-expression in double knockdown 3T3-L1 cells

elicited superior adipogenic rescue than TMEM120B, suggesting a dominant role of TMEM120A in adipogenesis. Intriguingly, Tmem120a is also upregulated by cold-exposure in WAT of mice²² and by treatment of 3T3-L1 mouse cells with rosiglitazone, an agonist of the master adipogenic transcription factor PPAR γ , that improves insulin sensitivity in diabetes²³."

Page 5, simplify the last sentence of paragraph 2. E.g., "accumulation, with hypothesized roles in nociception in brain." (?)

Thank you for this suggestion. Indeed, we struggled to explain these studies in a minimum of words and so have had to elaborate more in the text to accommodate the reviewer suggestion, which hopefully will make this clearer to all readers. This is the more important since the brain studies are completely unrelated. Moreover, in the time of our revision of our study, 3 papers have been published in *eLife* claiming the nociception study published in *Cell* to be completely incorrect. Therefore, we felt it important to separate and better clarify the studies. With these further clarifications, the ending of this paragraph now reads thus:

"Finally, brain is one of the few other tissues expressing low Tmem120a mRNA and antibodies detected a larger mass Tmem120a protein in mouse brain, suggesting a different isoform between adipose tissue and the nervous system²⁰. Tmem120A was one of the genes with most altered expression in hypothalamus of chickens genetically selected for fat accumulation²⁴. A different study suggested that TMEM120A (called there TACAN) is involved in nociception in brain²⁵; however, this presumed role as a mechanosensing channel was challenged by three independent studies published this year^{26,27,28}."

Page 5, edit and simplify the last two sentences for clarity.

We have now added minimal text to improve clarity here, changing it from:

"We find that loss of Tmem120a leads to a distinct but latent lipodystrophy pathology, similar to FPLD2, that we show is due to a novel disease pathomechanism: loss of Tmem120a alters positioning of multiple genes, enhancers, and miRNA-encoding loci resulting in multiple gene expression defects. This is the first demonstration of miRNA-encoding loci under NE positional regulation and it suggests that lipodystrophy-associated skeletal muscle hypertrophy could be paracrine mediated."

to:

"We find that loss of Tmem120a leads to a distinct latent lipodystrophy pathology, similar to FPLD2. At the cellular level, we find that loss of Tmem120a alters positioning of multiple genes, enhancers, and miRNA-encoding loci resulting in multiple gene expression defects, particularly reducing expression of fat metabolism genes and activating/derepressing muscle genes. This suggests a novel disease pathomechanism in which the altered genome organisation causes changes in many genes to affect adiposity. Importantly, this is the first demonstration that miRNA-encoding loci are under NE positional regulation and this, together with several targets of the altered miRNAs being upregulated muscle genes, suggests that lipodystrophy-associated skeletal muscle hypertrophy could be paracrine mediated."

Page 7, bottom paragraph, line 2: change "on HFD was accounted by reduced cell numbers (hyperplasia) or reduced fat accumulation (hypertrophy)" to "on HFD was due to fewer cells (should this be "hypoplasia", rather than "hyperplasia"?) or reduced fat accumulation per cell ("hypertrophy" seems incorrect here)"

Thank you for catching this. Indeed, the reviewer is correct and we meant "hypoplasia" and "atrophy" respectively. The offending sentence has been corrected.

Page 8, lines 2-3: I would not describe a 1-cm change in body length as "small"; indicate the percentage reduction. Do the reduced body and tail lengths warrant comment or speculation as a potential impact on skeletal development?

We have now added the % along with the length differences for both body and tail and removed the "small" designation for these results. We have no way of knowing without a separate study whether these effects are due to changes in circulating miRNAs or an indirect effect of the overall metabolism change or if it is specific to the fat stores in bone itself. Therefore, we do not think it appropriate to further elaborate on this observation in the text. We would note, however, that in the recently published study of a new lamin A lipodystrophy mouse model from the MacDougald lab (Corsa et al., 2021) they observed alterations in the fat within bone.

Page 9, second paragraph, first sentence is unclear ("constrained adipose expansion"?).

This is another phrase that might be only understood by the metabolism community, but it is nonetheless making an important distinction because the pathology is a failure to accumulate fat as opposed to a loss of pre-existing fat. Moreover, using terms like growth could be misinterpreted as indicating an expansion of cells that is not the case here. Therefore, we have had to expand this a bit for clarity from "*constrained adipose expansion*" to "*the relatively reduced accumulation of fat in individual adipocytes on HFD*".

Page 9, sentence that begins "Hyperlipidemia has been reported.." is overly long; needs editing for clarity.

This sentence is now divided into two as well as having been rewritten for clarity. It has been changed from:

"Hyperlipidemia has been reported in 42% of all lipodystrophy patients, most lamin-linked FPLD2 patients, and some of the wide spectrum of other lamin-linked lipodystrophies and metabolic syndromes^{30,31,32}; however, fasting triglyceride and free fatty acid levels were comparable between genotypes on both diets (Fig. 2g,h and Supplementary Table 1)."

to:

"A large study of multiple lipodystrophies reported hyperlipidemia in 42% of all patients; however, for lamin-linked lipodystrophies and metabolic syndromes different studies give widely different frequencies of hyperlipidemia, perhaps due to the small numbers of patients^{33,34,35}. Nonetheless, in our mice fasting triglyceride and free fatty acid levels were comparable between genotypes on both diets (Fig. 2g,h and Supplementary Table 1)."

Page 9, last sentence, define "polydipsia" and change "worsened" to "more severe".

We have added a clause to define polydipsia as "*excessive thirst observed in diabetes*" and changed "*worsened*" to "*more severe*".

Page 10, confusing first sentence (“To determine..”). E.g., to determine if Ad-Tmem120a-/- female mice accumulated less fat because they....., we performed...”?

For consistency with the above change from the “constrained adipose” usage and to explain more clearly what is being tested, we have changed this first sentence from:

“To determine whether altered substrate utilisation underpinned the constrained fat accumulation in Ad-Tmem120a^{-/-} female mice, we performed indirect calorimetry analysis.”

to:

“To determine whether the relatively reduced accumulation of fat in Ad-Tmem120a^{-/-} female mice on HFD might be related to a change in the carbon substrate used by mitochondria, we performed indirect calorimetry analysis.”

Page 15, middle: where is the data supporting this statement? “Tmem120a knockdown was 90% compared to..”.

The data was published in the Batrakou reference that was given at the end of this sentence. Notably, that is not just an experiment where we determined the knockdown for those conditions he had optimised, but the real data from the DamID experiment here because the PhD student at the time had done one large experiment where he also generated the material for the DamID. However, since he was close to graduating, we published a separate paper with some of the other data he had generated. Therefore, since the actual knockdown data was already published, we are just referencing it rather than including it in the figure.

We copy the data below from Figure 2A from that paper which shows that by antibody staining the Tmem120a protein was knocked down ~90% compared to non-target shRNA protein levels. We did not have antibodies to Tmem120b at the time, but we did test by qPCR, which revealed that Tmem120b was knocked down by >90% in Figure 4A from the same paper. Nonetheless, we thank the reviewer for raising this point because we wrote >90% in the text when in re-evaluating the data it was actually just very close to 90% and we have accordingly changed this in the text to “~90%”.

Supplementary Figures: If there are only 3 supplemental figures, why are they numbered 1, 3 and 5 instead of Supplemental Figures 1, 2 and 3?

Originally, we matched the numbers for Supplementary Figures with the number of the main text figure that it elaborated on. However, in this revision, we have now added several

Supplementary Figures that do not match to a main text figure in addressing reviewer comments and so they are now simply numbered consecutively.

Please find a list of the references for this Response to Reviewers document with DOI links below.

All The best,

Eric and Nik

Links for References:

Leibiger et al., 2013, <https://doi.org/10.1369%2F0022155413476868>

Wan et al., 2015, <https://doi.org/10.1007/s13577-015-0109-3>

Xue et al., 2021, <https://doi.org/10.7554/eLife.71220>

Rong et al., 2021, <https://doi.org/10.7554/eLife.71474>

Niu et al., 2021, <https://doi.org/10.7554/eLife.71188>

Corsa et al., 2021, <https://doi.org/10.2337/db20-1001>

Brown et al., 2016, <https://doi.org/10.1210/jc.2016-2466>

Meehan et. al., 2015, <https://doi.org/10.1586/17512433.2016.1096772>

REVIEWERS' COMMENTS

Reviewer #1 (Remarks to the Author):

This is a revision of the report regarding the generation of the Tmem120a knockout mice and the characterization of their latent lipodystrophy. Though there is much to be done to work out the mechanistic pathways suggested for phenotype development in mice and to prove their connection to human pathology, the authors are responsive and have addressed most of my concerns.

Minor changes:

Page 7, line 152, it should say (Supplementary Fig.1 h,i)

Reviewer #2 (Remarks to the Author):

The revised manuscript is outstanding, with previous concerns addressed fully by revision and in some cases additional data. This is a major contribution to understanding the mechanisms of 3D genome organization and metabolic disorders-- whether heritable or diet-related.

Text revisions below are required for accuracy or clarity:

Lines 374-375: change "different loci within the same gene accounted for ~20% (388) of the affected loci (Fig. 5g)." to "different genes within each locus (~20% [388] of affected loci; Fig. 5g)."

Line 398: change "they had failed to be recruited to the NE"
to "they had failed to be recruited to or maintained at the NE"

Lines 403-405: change "defect in the ability of the cells to respond to the challenge of the metabolic load of the HFD caused by the many metabolism and other genes altered in the knockout." to "defect in the ability of the knockout cells to respond to the challenge of the metabolic load of the HFD, due to the many metabolic and other genes altered in the knockout."

Line 411: change "we searched DRs for"
to "we searched DR genes for"

Line 489: change "in NE LMNA-linked FPLD2"
to "in FPLD2, caused by mutations in LMNA."

509-515: "Notably, these changes in the LMNA lipodystrophy patients paralleling the repositioning defects in our Tmem120a knockout mice are not due to loss of Tmem120a in the patients as levels were unchanged (Supplementary Fig. S6); however, due to extensive interactions of nuclear membrane proteins with the underlying lamina, the mutations in lamin A could still be expected to affect the gene repositioning function of Tmem120a."

to

"Since Tmem120a protein levels were unchanged in FPLD2 patients (Supplementary Fig. S6), we suggest FPLD2-mutated A-type lamins directly or indirectly affect the [localization or?] function of Tmem120a".

Lines 531-532: "interacts with many nuclear membrane proteins: indeed we previously demonstrated a probable interaction between Tmem120a and the lamin polymer20" doth protest too much. Stick to the data; e.g.: "interacts with many nuclear membrane proteins. Indeed, Tmem120a [co-

immunoprecipitates with A-type?] lamins from cells²⁰".

Line 746: change "sonicated and boiled for 5 min at 95°"
to "sonicated and heated 5 min at 95°".

Dear editors and reviewers,

Please find point by point response to final reviewers' comments. Editorial guidance is answered in detail in the table of the Author Checklist file.

Reviewer #1 (Remarks to the Author):

This is a revision of the report regarding the generation of the Tmem120a knockout mice and the characterization of their latent lipodystrophy. Though there is much to be done to work out the mechanistic pathways suggested for phenotype development in mice and to prove their connection to human pathology, the authors are responsive and have addressed most of my concerns.

We thank Reviewer 1 for accepting our revisions as sufficient.

Minor changes:

Page 7, line 152, it should say (Supplementary Fig.1 h,i)

The figure number is corrected now.

Reviewer #2 (Remarks to the Author):

The revised manuscript is outstanding, with previous concerns addressed fully by revision and in some cases additional data. This is a major contribution to understanding the mechanisms of 3D genome organization and metabolic disorders-- whether heritable or diet-related.

Thank you for the positive comment on our revisions.

Text revisions below are required for accuracy or clarity:

Lines 374-375: change "different loci within the same gene accounted for ~20% (388) of the affected loci (Fig. 5g)." to "different genes within each locus (~20% [388] of affected loci; Fig. 5g)."

The sentence is now corrected.

Line 398: change "they had failed to be recruited to the NE" to "they had failed to be recruited to or maintained at the NE"

The sentence is now corrected.

Lines 403-405: change "defect in the ability of the cells to respond to the challenge of the metabolic load of the HFD caused by the many metabolism and other genes altered in the knockout." to "defect in the ability of the knockout cells to respond to the challenge of the metabolic load of the HFD, due to the many metabolic and other genes altered in the knockout."

The sentence is now corrected.

Line 411: change "we searched DRs for" to "we searched DR genes for"

We believe that this sentence was misunderstood. The “Differential Regions (DR)” term reflects genomic regions that move between the nuclear envelope and nuclear interior as opposed to genes per se. Many DRs contain genes, but some do not. Within both populations there are some that contain enhancers. In this sentence we wanted to say that we specifically looked for predicted white adipose tissue enhancers (source: EnhancerAtlas) that were located within DR regions. Since the DR term was introduced far upstream and the discussion of genes in DR regions occurred between its introduction and the enhancer discussion, a reader could easily make this mistake. Therefore, we have tried to clarify this sentence by describing DRs again at this point, changing the sentence from:

“Therefore, we searched DRs for predicted enhancers, finding that a third of all pre-adipocyte enhancers that exhibited functional repression during adipogenesis were recruited to the NE while 31 enhancers that became activated in adipocytes were released from the NE (Fig. 6a).”

to:

“Therefore, we looked for predicted enhancers (source: Enhancer Atlas <http://www.enhanceratlas.org>) within genomic regions that are recruited to or released from the NE (i.e.: DR regions) during 3T3L1 differentiation. A third of the enhancers that are normally recruited to the NE failed to do so when Tmem120A was knocked down, while 31 enhancers that should be released from the NE remained at the NE upon Tmem120a knock-down (Fig. 6a).”

Line 489: change “in NE LMNA-linked FPLD2”
to “in FPLD2, caused by mutations in LMNA.”

The sentence is now corrected.

509-515: “Notably, these changes in the LMNA lipodystrophy patients paralleling the repositioning defects in our Tmem120a knockout mice are not due to loss of Tmem120a in the patients as levels were unchanged (Supplementary Fig. S6); however, due to extensive interactions of nuclear membrane proteins with the underlying lamina, the mutations in lamin A could still be expected to affect the gene repositioning function of Tmem120a.”

to

“Since Tmem120a protein levels were unchanged in FPLD2 patients (Supplementary Fig. S6), we suggest FPLD2-mutated A-type lamins directly or indirectly affect the [localization or?] function of Tmem120”.

The sentence is now corrected.

Lines 531-532: “interacts with many nuclear membrane proteins: indeed we previously demonstrated a probable interaction between Tmem120a and the lamin polymer” doth protest too much.

Stick to the data; e.g.: “interacts with many nuclear membrane proteins. Indeed, Tmem120a [co-immunoprecipitates with A-type?] lamins from cells”.

This sentence is now corrected from:

“...interacts with many nuclear membrane proteins: indeed we previously demonstrated a probable interaction between Tmem120a and the lamin polymer...”

to

“Indeed, it was previously published that Tmem120a is in the inner nuclear membrane and resists a pre-fixation detergent extraction that is characteristic of an interaction with the lamin polymer, and we have recently found that lamin A can be pulled down with Tmem120a and the two purified proteins expressed in bacteria can interact (unpublished data). The potential of TMEM120A to contribute to human adiposity disorders is further increased...”

Line 746: change “sonicated and boiled for 5 min at 95”
to “sonicated and heated 5 min at 95”.

The sentence is now corrected.

We hope that these corrections along with the amendments listed in Author Checklist file are sufficient and please let us know if more clarification is needed.

All the best,

Eric